RESEARCH

# Assembly of the threespine stickleback Y chromosome reveals convergent signatures of sex chromosome evolution

Catherine L. Peichel[1,2*], Shaugnessy R. McCann[1], Joseph A. Ross[1,3], Alice F. S. Naftaly[4], James R. Urton[1,3], Jennifer N. Cech[1,3], Jane Grimwood[5], Jeremy Schmutz[5], Richard M. Myers[5], David M. Kingsley[6] and Michael A. White[1,4*] ⓘD

* Correspondence: catherine.
peichel@iee.unibe.ch; whitem@uga.
edu
[1]Divisions of Human Biology and
Basic Sciences, Fred Hutchinson
Cancer Research Center, Seattle, WA
98109, USA
Full list of author information is
available at the end of the article

## Abstract

**Background:** Heteromorphic sex chromosomes have evolved repeatedly across diverse species. Suppression of recombination between X and Y chromosomes leads to degeneration of the Y chromosome. The progression of degeneration is not well understood, as complete sequence assemblies of heteromorphic Y chromosomes have only been generated across a handful of taxa with highly degenerate sex chromosomes. Here, we describe the assembly of the threespine stickleback (*Gasterosteus aculeatus*) Y chromosome, which is less than 26 million years old and at an intermediate stage of degeneration. Our previous work identified that the non-recombining region between the X and the Y spans approximately 17.5 Mb on the X chromosome.

**Results:** We combine long-read sequencing with a Hi-C-based proximity guided assembly to generate a 15.87 Mb assembly of the Y chromosome. Our assembly is concordant with cytogenetic maps and Sanger sequences of over 90 Y chromosome BAC clones. We find three evolutionary strata on the Y chromosome, consistent with the three inversions identified by our previous cytogenetic analyses. The threespine stickleback Y shows convergence with more degenerate sex chromosomes in the retention of haploinsufficient genes and the accumulation of genes with testis-biased expression, many of which are recent duplicates. However, we find no evidence for large amplicons identified in other sex chromosome systems. We also report an excellent candidate for the master sex-determination gene: a translocated copy of Amh (*Amhy*).

**Conclusions:** Together, our work shows that the evolutionary forces shaping sex chromosomes can cause relatively rapid changes in the overall genetic architecture of Y chromosomes.

**Keywords:** Threespine stickleback, Y chromosome, Gene duplication, Sex determination

## Background

Heteromorphic and highly degenerate sex chromosome pairs (i.e., XY or ZW) can ultimately evolve from autosomal ancestors when recombination is suppressed between them (reviewed in [1]). Thus, sex chromosomes are an intriguing region of the genome to understand how mutations and repetitive DNA accumulate in the absence of recombination and how gene content evolves once a chromosome becomes sex-limited. Although this degenerative process was originally assumed to occur at a constant rate, recent work has revealed that the tempo and outcome of sex chromosome degeneration is highly variable and not always correlated with the age of the sex chromosome [2, 3]. Thus, understanding the dynamics of sex chromosome evolution requires characterizing systems at different stages of degeneration [1, 4]. At the early stages of evolution, sex chromosomes are expected to be mostly homomorphic at the cytogenetic level, with very few sequence differences between the proto-X and Y. At intermediate stages, suppression of recombination spreads, possibly due to selection for linkage between the sex determination gene and additional sexually antagonistic mutations, resulting in additional sequence divergence, loss of genes, accumulation of transposable elements, and rearrangements like inversions. At this intermediate stage, visibly heteromorphic sex chromosomes can emerge [1, 4, 5]. The last stages of this degenerative process, resulting in highly heteromorphic sex chromosomes, were once predicted to be an evolutionary dead end, with the Y (and W) chromosomes inevitably losing functional gene copies across the entire chromosome as deleterious mutations accumulate [6]. Contrary to this expectation, assembly of multiple mammalian Y chromosome sequences [7–12], the chicken W chromosome [13], and invertebrate Y chromosomes [14, 15] has revealed that the sequence of the sex-limited chromosome is much more dynamic, punctuated by gene gains and losses, rather than becoming entirely degenerated.

Although short-read sequencing of sex chromosomes has yielded insight into how ancestral single-copy genes have evolved between X and Y chromosomes (e.g., [16–18]), these approaches cannot be used to study how Y chromosomes have structurally evolved once they become heteromorphic. Short-reads cannot span many of the lengthy repeat units' characteristic of Y chromosomes, leading to a collapse of these regions during the assembly process. Because of the inherent difficulty in assembling these highly repetitive regions of the genome, heteromorphic Y chromosomes have been omitted from many reference genome assemblies. Most of these existing reference Y chromosome assemblies were constructed through labor intensive, iterative Sanger sequencing of large inserts from bacterial artificial chromosome (BAC) libraries [7–9, 11]. However, recent technological advances, such as Pacific Biosciences (PacBio) long-read sequencing, chromatin interaction maps, and optical mapping, have enabled the assembly of additional heteromorphic Y chromosomes [12, 14, 15].

Through these assemblies, two classes of genes have been identified on highly degenerate and relatively old sex chromosomes. The first are dosage-sensitive genes that were present in the common ancestor of both chromosomes and have been maintained as single copies on the Y chromosome across multiple mammalian lineages [10, 19] as well as on the degenerating W chromosome of birds [13]. The second are genes that exist in high copy number families on Y chromosomes and generally have gene expression patterns restricted to the testes, suggesting roles in spermatogenesis [7, 8,

11, 20–22]. It is clear that the genetic architecture of sex chromosomes can be shaped by multiple processes over long evolutionary time scales. However, reference assemblies of sex chromosomes that are already heteromorphic but at an earlier stage of degeneration are largely absent, with the exception of the young neo-Y chromosome assembly of *Drosophila miranda* [14, 23], making it unclear whether the genetic architecture of evolving sex chromosomes is shaped by these evolutionary forces at earlier stages.

The threespine stickleback fish (*Gasterosteus aculeatus*) is an excellent model system to explore the structural evolution of sex chromosomes. Although the threespine stickleback has a high-quality reference genome assembly [24] that has gone through multiple iterations of refinement [25–27], the assembly was derived from a female fish, precluding the Y chromosome from assembly. The threespine stickleback has a heteromorphic XY sex chromosome system that is shared across the *Gasterosteus* genus but not with other species in the Gasterosteidae family and therefore evolved between 14 and 26 million years (14–26 million generations) ago [28–31]. This Y chromosome is younger than the highly degenerate Y chromosome of mammals that evolved ~ 180 million years ago [10, 19] and appears to be at an earlier stage of degeneration. Crossing over is suppressed between the X and Y chromosomes over a majority of their length, resulting in an approximately 2.5 Mb pseudoautosomal region of the 20.6 Mb X chromosome [25]. The region of suppressed crossing over is coincident with three pericentric inversions that differentiate the X and Y chromosomes [32]. Illumina-based sequencing suggested the non-crossover region on the Y chromosome was composed of two differently aged evolutionary strata, the oldest of which retained genes that were predicted to be haploinsufficient [18]. However, all studies in threespine stickleback have relied on mapping short-reads to the reference X chromosome, limiting our understanding to regions conserved between the X and Y. It has not yet been possible to explore how unique structure and sequence is evolving across this heteromorphic Y chromosome.

Here, we report a high-quality reference assembly of a vertebrate Y chromosome at an intermediate stage of degeneration. We combined high-coverage, long-read PacBio sequencing with chromatin conformation capture sequencing (Hi-C) to assemble a full scaffold of the threespine stickleback Y chromosome. Our assembly is completely concordant with more than 90 Sanger-sequenced inserts from a bacterial artificial chromosome (BAC) library and with a known cytogenetic map [32]. Throughout the male-specific region, we have identified several novel sequence and structural characteristics that parallel patterns observed on highly degenerate sex chromosome systems. The sex chromosome of threespine stickleback is a useful model system to understand the step-wise evolution of the genetic architecture on sex-limited chromosomes.

## Results

### De novo assembly of the threespine stickleback Y chromosome

We used high-coverage PacBio long-read sequencing to assemble a threespine stickleback genome from a male fish of the Paxton Lake Benthic population (British Columbia, Canada). Raw read coverage was approximately 75.25x across the genome (34.84 Gb total sequence) (Additional file 1: Table S1). The longest raw PacBio reads

were assembled using the Canu pipeline, refined by Arrow, resulting in a primary contig assembly of 622.30 Mb across 3593 contigs (Additional file 1: Table S1). This assembly size was considerably larger than the Hi-C revised threespine stickleback female genome assembly (463.04 Mb including autosomes and X chromosome) [24, 26, 27]. The increased assembly length was largely due to heterozygous loci being separated into individual alleles (haplotigs). Of the total Canu assembly, 3134 contigs (574.67 Mb) aligned to 442.41 Mb of autosomes in the reference assembly. Only 129 contigs partially aligned to the genome (less than 25% of the contig length aligned; 10.15 Mb) and 148 contigs did not align at all to the genome (3.58 Mb). We collapsed 118.89 Mb of haplotigs, reducing the 574.67 Mb alignment to 455.78 Mb of non-redundant sequence across the autosomes, an estimate closer to the 442.41 Mb of autosomes in the female reference genome assembly.

We targeted Y-linked contigs in the Canu assembly by identifying contigs that shared reduced sequence homology with the reference X chromosome or did not align to the autosomes. In the youngest region of the threespine stickleback sex chromosomes (the previously identified stratum two), the X and Y chromosomes still share considerable sequence homology. However, within this stratum, heterozygosity is even higher than what is observed across the autosomes [18]. Based on this divergence, Canu should separate X- and Y-linked contigs during the initial assembly process. Contigs aligned to the X chromosome formed a distribution of sequence identity that was not unimodal, reflecting the presence of both X- and Y-linked contigs (Additional file 2: Fig. S1). Setting a sequence identity threshold of 96% resulted in a set of 114 X-linked contigs that totaled 21.26 Mb, compared to the previous 20.62 Mb X chromosome reference assembly. There were 68 putative Y-linked contigs that had a sequence identity less than or equal to 96%, totaling 12.64 Mb. The oldest region of the Y chromosome (stratum 1) contains many regions that have either been deleted or diverged to such an extent that sequencing reads cannot be mapped to this region [18]. Consequently, there may be contigs unique to the Y chromosome that cannot be captured through alignments to the reference X chromosome. To account for these loci, we also included the contigs that only partially aligned to the genome (less than 25% of the contig length aligned; 129 contigs; 10.15 Mb) or did not align at all to the genome (148 contigs; 3.58 Mb) in the set of putative Y-linked contigs (345 total contigs).

### Hi-C proximity-guided assembly yielded contiguous scaffolds of the sex chromosomes

We used chromosome conformation capture (Hi-C) sequencing and a proximity-guided method to assemble the set of putative X- and Y-linked contigs into scaffolds. Using the 3D-DNA assembler [33], 105 of the 114 X-linked contigs were combined into three main scaffolds that totaled 20.78 Mb. The scaffolds were largely colinear with the reference X chromosome, with scaffolds one and two aligning to the pseudoautosomal region and scaffold three mostly aligning to the remainder of the X chromosome that does not recombine with the Y (Additional file 2: Fig. S2).

We assembled the putative Y-linked contigs using the same process. Of the 345 total contigs, 115 were initially combined into a single primary scaffold that totaled 17.15 Mb. We visually inspected the Hi-C interaction map for any sign of misassembled contigs. There was a clear mis-joining of contigs near one end of the primary scaffold,

where there were fewer short-range Hi-C interactions at the diagonal combined with an overall absence of long-range Hi-C interactions between all of the contigs in this region and the remainder of the Y scaffold (Additional file 2: Fig. S3). We manually removed this cluster of contigs from the primary scaffold (45 contigs; 1.86 Mb), resulting in an initial Y chromosome scaffold totaling 15.28 Mb across 70 contigs (Fig. 1a).

## Bacterial artificial chromosome library sequences are concordant with the assembled Y chromosome

To assess the overall accuracy of our assembly, we compared our assembly to Sanger sequenced inserts from a bacterial artificial chromosome (BAC) library constructed from males from the same population. Mean insert size among the 101 sequenced BAC clones was 168.13 kb, similar in size to the average contig length within the Y chromosome scaffolds (217.85 kb). Using the BAC sequences, we were able to identify whether any of the contigs within the scaffold contain collapsed haplotigs between the X and Y

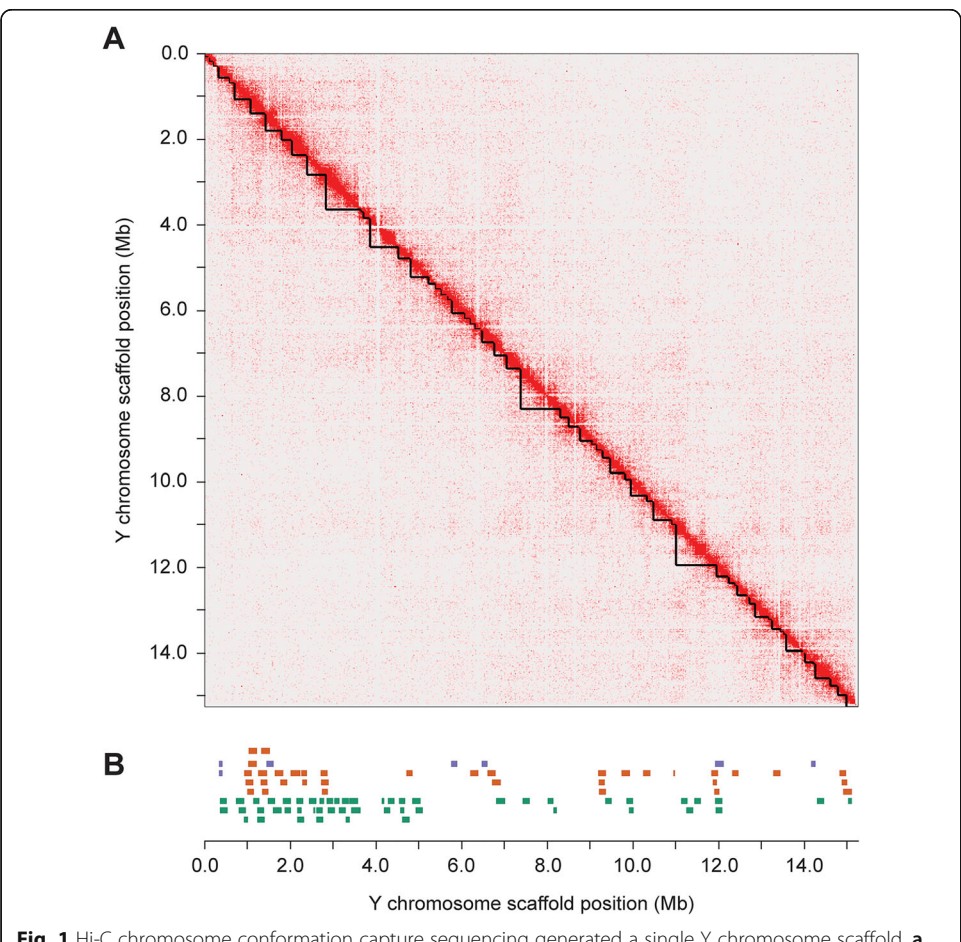

**Fig. 1** Hi-C chromosome conformation capture sequencing generated a single Y chromosome scaffold. **a** The contact matrix shows an enrichment of interactions between contigs in close proximity along the diagonal. Contig boundaries in the assembly are denoted by the black triangles along the diagonal. **b** Sanger sequenced BAC inserts that align concordantly throughout the scaffold are shown, with BACs that spanned gaps between contigs in orange, BACs that extended into, but did not span gaps in purple, and BACs that were contained completely within an individual contig in green

chromosomes (chimeric contigs should contain reduced sequence identity when aligned to known Y chromosome BAC contigs). In addition, the contig ordering across the scaffold was verified by BAC contig sequences that spanned gaps in the assembly. We aligned all 101 sequenced BAC contigs to the Y chromosome scaffold and found 92 of the BAC contigs aligned concordantly with the assembly (Fig. 1b). These BACs aligned to 40 of the 70 contigs in the assembly with a high sequence identity (7.72 Mb of non-overlapping sequence in the 15.28 Mb assembly aligned concordantly to the BAC contigs). The remaining 9 BAC contigs that did not align concordantly indicate there are small-scale structural differences between the Canu Y chromosome assembly and the BAC clones derived from a separate Paxton Lake male threespine stickleback, either reflecting errors in the Y chromosome assembly, rearrangements in the BAC clone sequences, or true polymorphisms segregating in the Paxton Lake benthic population. Four of the discordant BACs aligned to regions of the reference Y that were greater than the Sanger sequenced length of the BAC insert, suggesting possible indels. The remaining five discordant BACs contained sub alignments with mixed orientations, suggesting possible small-scale inversions not present in our assembly.

Among the aligned BAC contigs, many provided additional sequence information, either spanning gaps between contigs in the Y chromosome assembly or extending from contigs into gaps in the assembly. Of the 92 BAC contigs that aligned concordantly, seven BAC contigs extended into five different gaps in the assembly and 35 BAC contigs spanned 18 different gaps in the assembly (26% of the total gaps in the assembly) (Fig. 1b). The remainder of the aligned BAC contigs aligned completely within an individual contig in the Y assembly. We merged this additional sequence into the initial Y chromosome assembly, resulting in a merged Y chromosome scaffold that contained 52 contigs, totaling 15.78 Mb.

### The Y chromosome assembly is concordant with known cytogenetic maps

The threespine stickleback Y chromosome has undergone at least three pericentric inversions relative to the X chromosome, forming a non-crossover region that spans a majority of the chromosome [32]. These inversions were mapped by ordering a series of cytogenetic markers along both the X and Y chromosomes (Fig. 2a). To determine whether our Y chromosome assembly was consistent with the known cytogenetic marker ordering, we used BLAST to locate the position of each marker within the assembly. We were able to locate four of the five markers used from the male-specific region in our assembly. The position of these cytogenetic markers was concordant with our assembly (Fig 2b). The missing marker in the non-crossover region (*STN235*) likely reflects a region of our Y reference that is not fully assembled or is a true deletion within the Paxton Lake benthic population, relative to the Pacific Ocean marine population used for the cytogenetic map [32].

The location of the oldest region within the Y chromosome (the previously identified stratum one) had been ambiguous. Cytogenetic markers from this region could not be hybridized to the Y chromosome [32], suggesting this region may be largely deleted or highly degenerated. Subsequent work using Illumina short-read sequencing revealed that some genes from this region were still present on the Y chromosome under strong purifying selection, but the location of these genes within the Y could not be

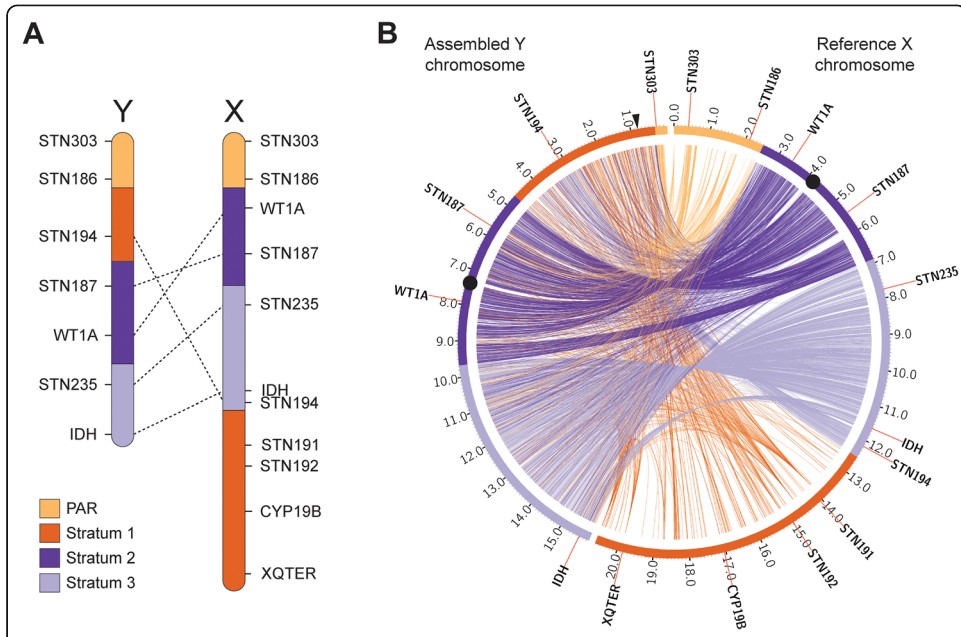

**Fig. 2** The threespine stickleback Y chromosome assembly is concordant with cytogenetic maps. **a** The Y chromosome has diverged from the X chromosome through a series of inversions determined through ordering of cytogenetic markers (dashed lines indicate rearrangements of the linear order of markers [32]). **b** Alignments of the assembled Y chromosome (left) to the X chromosome (right) reveal the same inversions. A majority of the pseudoautosomal region is not included in the reference Y chromosome assembly because this region was not targeted (see methods). The location of the candidate sex determination gene *(Amhy)* is indicated by the black arrow. Centromeres are shown by black circles. Positions are shown in megabases

determined by mapping reads to the X chromosome [18]. The cytogenetic marker, *Idh*, is located at the distal end of our Y chromosome assembly, remarkably consistent with the placement of *Idh* in the cytogenetic map [32], indicating stratum one is no longer located at the distal end of the Y chromosome as it is on the X chromosome. Instead, we found a high density of stratum one alignments near the boundary of the pseudoautosomal region at the opposite end of the chromosome (Fig. 2b). Within this stratum, there was an overall lower density of alignments between the X and Y chromosomes, consistent with previous patterns mapping Illumina short reads to the reference X chromosome [18]. The placement of stratum one in the assembly was consistent with the hybridization of a fluorescent in situ hybridization probe, designed from a stratum one BAC insert. This probe clearly hybridized to the chromosome end opposite of *Idh* (Additional file 2: Fig. S4).

Because we were primarily focused on sequences that were highly divergent from the X chromosome or absent from the female reference genome entirely, our strategy did not target the pseudoautosomal region for assembly into the Y chromosome. Nevertheless, our assembly did place a small fraction of the ~ 2.5 Mb pseudoautosomal region on the distal end of the male-specific Y chromosome, adjacent to stratum one. The cytogenetic marker *STN303* was included in this region, which is located on the opposite end of the pseudoautosomal region on the X chromosome (Fig. 2). This discordance in marker placement within the pseudoautosomal region likely indicates a misassembly of the region. The pseudoautosomal region contains repetitive sequence, complicating overall assembly of the region (see transposable elements section). Indeed,

the contigs spanning this region and *STN303* have a smaller size (five contigs; median, 88,098 bp) than the remaining contigs within the Y chromosome or X chromosome, consistent with highly heterozygous, repetitive sequence.

### Short-read sequencing correctly aligns to the X and Y chromosomes

We aligned Illumina short-read sequencing from males and females of three different threespine stickleback populations in order to test whether the inclusion of a reference Y chromosome allows correct alignment of X- and Y-linked short-reads or if there are regions of the sex chromosomes where short-reads cross align. These regions could indicate highly homologous sequence between the X and Y chromosomes or chimeric assembly errors. At a fine-scale, we observed some variation in read depth across the X and Y chromosomes, which may indicate some cross-aligning of short-reads between the sex chromosomes (Additional file 2: Fig. S5 and Fig. S6). These small regions may reflect chimeras in the assembly or may reflect accurately assembled regions that are highly homologous between the X and Y chromosomes. Indeed, these small regions were mostly located in the younger strata of the sex chromosomes (see below). However, our assembly is highly accurate overall and can correctly align X- and Y-linked reads. Male reads exhibited a median 0.5x coverage relative to autosomes on both the X and Y chromosomes. In addition, female reads had median coverages of 0.0x on the Y chromosome and 1.0x on the X chromosome, relative to autosomes (Additional file 2: Fig. S7).

### The location of centromeric repeats are concordant with a metacentric chromosome

A 186 bp centromeric AT-rich alpha satellite repeat was previously identified in female fish by chromatin immunoprecipitation followed by sequencing (ChIP-seq) [34]. Although this repeat hybridized strongly to autosomes and the X chromosome, there was only weak hybridization of the probe to the Y chromosome, suggesting the Y chromosome might have a divergent centromeric repeat and/or contain substantially less satellite DNA than the autosomes [34]. We used ChIP-seq with the same antibody against centromere protein A (CENP-A) in males to identify any Y chromosome repeats. Relative to the input DNA, we found strong enrichment of reads from the immunoprecipitation mapping to the center of the Y chromosome assembly, indicative of CENP-A binding (Fig. 3a; Additional file 2: Fig. S8). The enrichment was located between cytogenetic markers *STN187* and *WT1A*, consistent with the predicted location of the centromere in the cytogenetic map and the metacentric chromosome morphology in karyotypes [32]. These results further confirm the ordering of contigs within our Y chromosome scaffold.

Underlying the CENP-A peak, we found a core centromere AT-rich repeat. We identified 14 copies of the repeat in our Y chromosome assembly, which shared an average pairwise sequence identity of 84.6% with the core repeat that hybridized to the remainder of the genome [34] (Additional file 2: Fig. S9). The repeats fell at the edges of a gap, indicating that a majority of the repeats were not assembled into our primary scaffold. Uneven coverage signal in Hi-C libraries from repetitive DNA can trigger the 3D-DNA assembler to remove these regions from contigs during the editing step [33, 35]. Consistent with this, both contigs that flanked the centromere gap in the Y

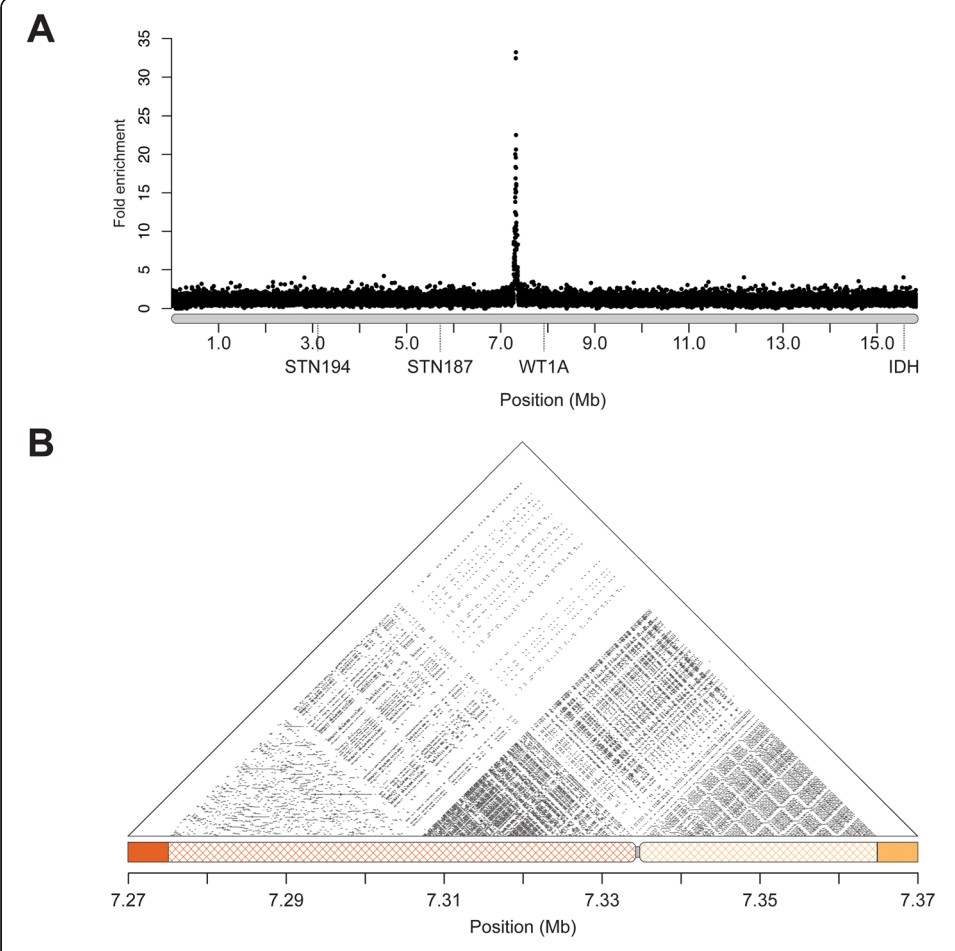

**Fig. 3** Sequences immunoprecipitated with CENP-A are enriched at the center of the chromosome. **a** Short-read sequences from a chromatin-immunoprecipitation (ChIP-seq) with CENP-A were aligned to the reference Y chromosome assembly. There is a prominent peak between markers STN187 and WT1A where the centromere is located. CENP-A enrichment from a second male fish is shown in (Additional file 2: Fig. S8). **b** Alpha satellite monomeric repeats are organized into higher order repeats (HORs). Sequence identity is shown in 100 bp windows across the centromere sequence of the Y chromosome. 87 kb of sequence containing the monomeric repeat was rejoined (crosshatched) to contigs that were previously fragmented in the scaffolding process (orange contig: 11,894; yellow contig: 11,839). The gap between the two contigs is shown in gray

chromosome assembly had additional sequence that was removed by the 3D-DNA pipeline as "debris." The first contig that was adjacent to the gap (contig 11,894) contained six copies of the repeat and had an additional 57,692 bp that was removed as "debris." The second contig on the opposite side of the gap (contig 11,839) had eight copies of the repeat and an additional 29,308 bp of sequence that was removed as "debris." We used BLAST to search for additional repeats in the debris using the majority consensus sequence of the 14 previously identified centromere repeats in the Y assembly. There were an additional 304 repeats in the debris sequence from contig 11,894, and 163 repeats in the debris sequence from contig 11,839. We added the debris sequence back into the total Y chromosome assembly, increasing the assembled centromere size by 87 kb (total Y chromosome length: 15.87 Mb) (Fig. 3b). Average pairwise percent sequence identity among all monomeric repeats in the Y chromosome assembly was 89.5%. Compared to the core threespine stickleback centromere repeat

previously identified, the Y chromosome centromere repeat was more divergent. Average pairwise percent sequence identity between all the motifs in the Y chromosome assembly and the centromere repeat identified from female fish was only 86.8%.

   Centromeres are often composed of highly similar blocks of monomeric repeats, organized into higher order repeats (HORs) [36–38]. Previous characterization of the monomeric centromere repeat in threespine stickleback did not reveal a HOR organization; however, this analysis was limited by the identification of only a few short stretches of the monomeric repeat on each autosome [34]. The ~ 87 kb of assembled centromere on the Y shows a clear higher order patterning around the centromeric region, consistent with complex HORs (Fig. 3b).

### The Y chromosome has three evolutionary strata

Previous estimates of synonymous site divergence ($d_S$) in coding regions have indicated there are two evolutionary strata on the threespine stickleback sex chromosomes [18], despite the presence of at least three major inversion events in the cytogenetic map of the sex chromosomes [32]. Because these estimates relied on aligning short-read Illumina sequences to the reference X chromosomes, overall divergence could have been biased by mapping artifacts, especially in the oldest region of the Y chromosome. We investigated whether our Y chromosome assembly supported the earlier model of two evolutionary strata or whether there could be additional strata uncovered in the current assembly. We aligned all Ensembl predicted X chromosome coding regions outside of the pseudoautosomal region to the Y chromosome reference assembly to estimate divergence. Of the 1184 annotated coding sequences, we were able to align 522 (44.1%) to the male-specific region of the Y chromosome (Table 1). We found a clear signature of three evolutionary strata, consistent with inversion breakpoints within the cytogenetic map as well as within our de novo reference assembly. The oldest stratum (stratum one) encompassed the same region of the X chromosome as previously described in the Illumina-based study and had highly elevated $d_S$ (stratum one median $d_S$, 0.155). In contrast to the Illumina-based estimates, our new assembly revealed that the remainder of coding regions across the X chromosome formed two distinct strata, with different estimates of $d_S$ (Fig. 4; Table 1). We also investigated whether the older strata had increased non-synonymous divergence ($d_N$) consistent with inefficient selection from the lack of crossing over between the chromosomes [39, 40]. As predicted, stratum one had a significantly higher $d_N$ than strata two and three (Table 1). Stratum two had a significantly lower $d_N$ than the other strata. This was also reflected by a significantly lower $d_N/d_S$ ratio (Table 1), suggesting genes in stratum two are under stronger purifying selection.

**Table 1** Median nucleotide divergence between X and Y chromosome gametologs

|  | X-linked | Y-linked | Percent remaining on Y | $d_S$ | $d_N$ | $d_N/d_S$ |
|---|---|---|---|---|---|---|
| Stratum 1 | 610 | 110 | 18.0% | 0.155[a] | 0.030[a] | 0.287[a] |
| Stratum 2 | 242 | 183 | 75.6% | 0.042[b] | 0.009[b] | 0.203[b] |
| Stratum 3 | 332 | 229 | 69.0% | 0.033[c] | 0.012[c] | 0.341[a] |

[a,b,c]Groups significantly different by a pairwise Mann-Whitney $U$ test; $P < 0.05$

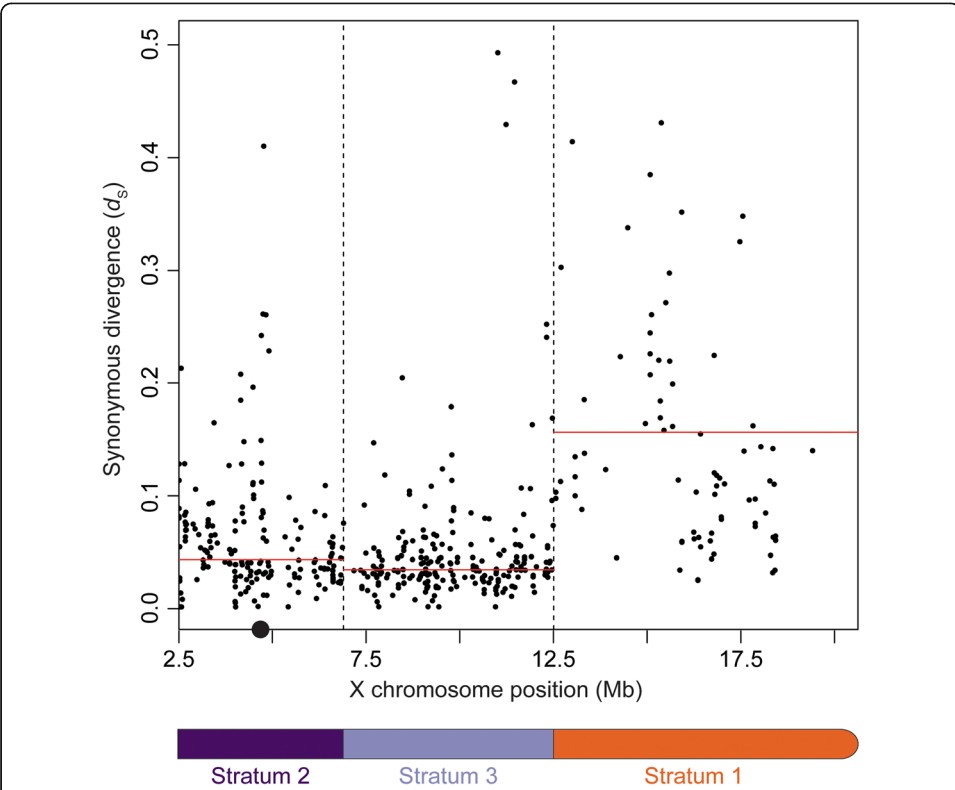

**Fig. 4** The sex chromosomes have three distinct evolutionary strata. Synonymous divergence ($d_S$) between the X and Y chromosome was estimated for every annotated transcript on the X chromosome. Genes are ordered by position on the X chromosome (Mb). Median divergence across each stratum is shown by the red line; values are given in Table 1. Strata breakpoints are indicated by the vertical dashed lines. The centromere is indicated by a black circle. The pseudoautosomal region (positions before 2.5 Mb) is not shown. Genes with $d_S$ divergence above 0.5 are not shown

We verified that the increased divergence we observed between the X and Y chromosomes was not driven by sequencing error in the long-read assembly. We aligned short-read Illumina sequences from three populations of threespine stickleback fish, including an independent male from the Paxton Lake benthic population from which the reference Y chromosome assembly was assembled. In all cases, nucleotide divergence was an order of magnitude lower on the Y chromosome compared to the X chromosome or the autosomes (Additional file 1: Table S2). The lowest divergence we observed ($2.5 \times 10^{-5}$ substitutions per site) was between the reference Y chromosome and an Illumina-sequenced Y chromosome from the same population. Combined, these results indicate our long-read PacBio assembly has a low sequencing error rate and reveal that the Y chromosome has reduced sequence diversity within threespine stickleback fish, relative to the remainder of the genome.

## The Y chromosome is evolving a unique genetic architecture

Haploinsufficient genes have been repeatedly retained on degenerating sex chromosomes of mammals and birds [10, 13] and may be enriched in stratum one of the stickleback Y chromosome [18]. We explored whether our expanded set of annotated genes exhibited signatures of haploinsufficiency by identifying orthologs between the

X-annotated genes and human genes ranked for haploinsufficiency (Decipher Haploin-sufficiency Predictions (DHP) v. 3) [41, 42]. Within strata one and two, we found genes with a retained Y-linked gametolog had lower DHP scores than genes without a Y-linked gametolog, indicating that retained genes were more likely to exhibit haploinsufficiency (Fig. 5; Mann-Whitney *U* test; stratum one *P* < 0.001; stratum two *P* = 0.035). We found a similar trend for genes retained on the Y chromosome in stratum three, but this result was not significant (Fig. 5; Mann-Whitney *U* test; *P* = 0.085). Nevertheless, this lower score suggests enrichment for haploinsufficient genes may already be underway within the youngest region of the Y chromosome.

Genes can be acquired on the Y chromosome through duplications from autosomes (reviewed in [43]), a process that has had a prominent impact on the overall gene content of highly degenerate sex chromosomes [7, 44–49], but the overall influence of this process on the genetic architecture of sex chromosomes at earlier stages of degeneration has not been documented. To identify whether the stickleback Y chromosome also contained genes shared with autosomes, but not the X chromosome, we first used the MAKER gene annotation pipeline [50, 51] to assemble a complete set of coding regions across the Y chromosome reference sequence. We identified a total of 626 genes across the male-specific region of the Y chromosome, 33 of which had paralogs

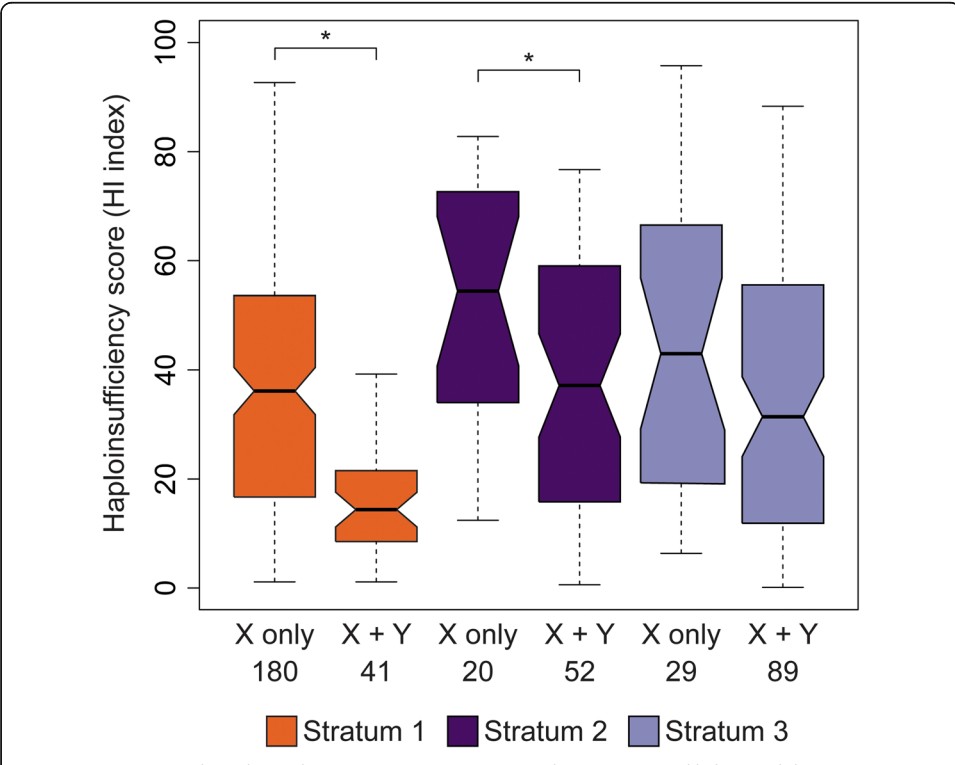

**Fig. 5** Genes retained on the Y chromosome in strata one and two are more likely to exhibit haploinsufficiency. Human proteins with predicted haploinsufficiency indexes were matched to one-to-one human-threespine stickleback fish orthologs from the X chromosome. Haploinsufficiency indexes were significantly lower for genes retained on both the X and Y chromosomes than for genes present only on the X chromosome (i.e., lost from the Y chromosome) in both strata one and two. A lower index indicates that a gene is more likely to be haploinsufficient. The total number of genes in each category is shown. The median is indicated by the solid black line. Whiskers denote 1.5x the interquartile range. Outliers are not shown. Asterisks indicate *P* < 0.05 (Mann-Whitney *U* test)

on autosomes, but not on the X chromosome (5.3%) (Table 2). A majority of these genes (25 of 33; 75.8%) appeared to have undergone duplications within the Y chromosome following translocation from the autosomes (genes had copy numbers ranging from two to six). Gene translocation onto sex chromosomes can occur through RNA-mediated mechanisms (retrogenes) or through DNA-based translocations (reviewed in [52]). Of the stickleback genes that had multiple introns within the autosomal paralog (31 of 33 genes), we did not detect a single paralog on the Y chromosome that had a complete loss of introns.

Genes that accumulate on Y chromosomes are predicted to have male beneficial functions. On many highly degenerate sex chromosomes, genes that have translocated to the Y chromosome from autosomes exhibit testis-biased expression [7, 44, 45], suggesting important roles in spermatogenesis. To determine whether the translocated genes on the threespine stickleback Y chromosome have testis-biased gene expression relative to the single-copy ancestral genes, we compared expression between testis tissue and three other tissues (liver, brain, and larvae). Compared with all tissues, we found stronger testis-biased expression among the genes that translocated to the Y chromosome, compared to the single-copy genes with a gametolog on the X chromosome (Fig. 6; Mann-Whitney $U$ test; $P < 0.05$). Because DNA-based translocations of genes often contain their native regulatory elements, we examined whether the autosomal paralogs also exhibited testis-biased expression to a similar degree as the Y-linked paralogs. Consistent with this pattern, we observed a similar degree of testis-biased expression between testis and liver tissue among the ancestral paralogs on the autosomes (median translocated genes $Log_2$ fold change: $- 0.867$; median ancestral autosomal paralog $Log_2$ fold change: $- 1.558$; Mann-Whitney $U$ test, $P = 0.818$). This pattern did not hold for comparisons between testis and larvae (median translocated genes $Log_2$ fold change, $- 5.178$; median ancestral autosomal paralog $Log_2$ fold change, $- 1.371$; Mann-Whitney $U$ test $P < 0.001$) and testis and brain (median translocated genes $Log_2$ fold change, $- 3.548$; median ancestral autosomal paralog $Log_2$ fold change, $- 1.601$; Mann-Whitney $U$ test $P = 0.036$). Combined, our results indicate that the genes which translocated to the Y chromosome and were retained often had testis-biased expression ancestrally.

Duplicated genes on the Y chromosome can also be derived from ancestral genes shared between the X and Y. Of the 626 genes annotated across the male-specific region of the Y chromosome, 47 (7.5%) had greater than one copy on the Y chromosome and also had an X-linked gametolog (Table 2). None of these genes

**Table 2** Origin of genes in each stratum on the Y chromosome

|  | X ancestral, Y single copy | X ancestral, Y duplicated | Autosomal, Y single copy | Autosomal, Y duplicated | Unknown origin, Y single copy | Unknown origin, Y duplicated | Total |
|---|---|---|---|---|---|---|---|
| Stratum 1 | 114 (72.6%) | 14 (9.0%) | 3 (1.9%) | 6 (3.8%) | 19 (12.1%) | 1 (0.6%) | 157 |
| Stratum 2 | 154 (80.6%) | 11 (5.8%) | 2 (1.0%) | 11 (5.8%) | 11 (5.8%) | 2 (1.0%) | 191 |
| Stratum 3 | 233 (83.8%) | 22 (7.9%) | 3 (1.1%) | 8 (2.9%) | 11 (4.0%) | 1 (0.3%) | 278 |

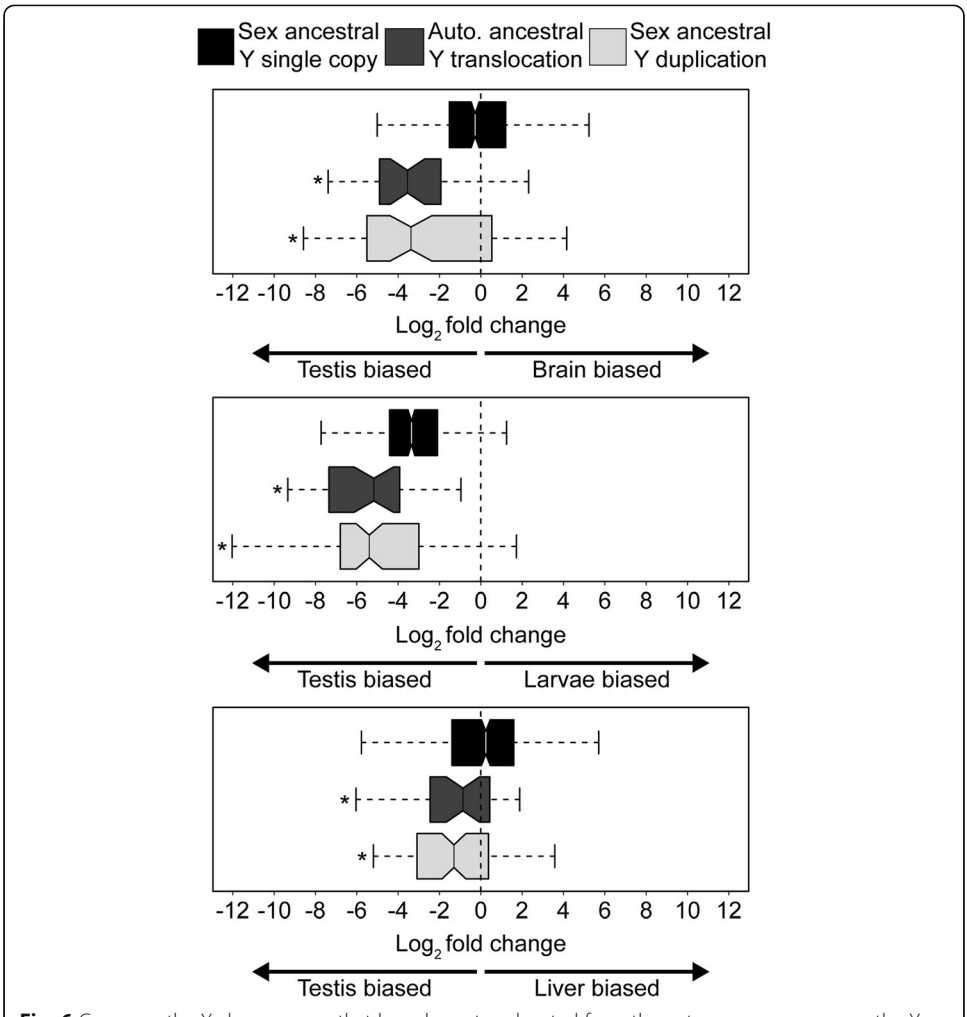

**Fig. 6** Genes on the Y chromosome that have been translocated from the autosomes or genes on the Y chromosome that have been duplicated show testis-biased gene expression. Log₂ fold change between testis tissue and three other tissues (brain, larvae, and liver) is shown. The median is indicated by the solid line. Whiskers denote 1.5x the interquartile range. Outliers are not shown. For each tissue comparison, asterisks denote groups with significantly different expression from single-copy genes on the Y chromosome that have an X-linked gametolog (Sex ancestral; Mann-Whitney $U$ test; $P < 0.05$)

were structured within large amplicons, which are characteristic of many mammalian Y chromosomes [7–9, 11, 22, 53–55]. Instead, copy number ranged from two to seven copies total. We explored if this duplicated class of genes also exhibited testis-biased expression similar to what we observed with the autosome translocated genes. Consistent with the previous patterns, we found strong testis-biased expression between testis and all other tissues among duplicated genes that have an X-linked gametolog (Fig. 6; Mann-Whitney $U$ test; $P < 0.001$, all comparisons). Similar to the ancestral autosomal paralogs, we found that genes often exhibited testis-biased expression ancestrally on the X chromosome before duplicating on the Y (Additional file 1: Table S3). However, this pattern did not hold in all tissue comparisons. In some cases, genes exhibited stronger testis-biased expression after duplicating on the Y chromosome.

## Transposable elements have accumulated throughout the Y chromosome

Transposable elements also rapidly accumulate on sex chromosomes once recombination is suppressed (reviewed in [1]). The threespine stickleback Y chromosome has a higher density of transposable elements throughout the male-specific region of the Y chromosome, compared to the X chromosome (Additional file 2: Fig. S10 and Fig. S11). We found the highest densities within stratum one, consistent with recombination being suppressed in this region for the greatest amount of time. We also found a slightly higher density of transposable elements within the pseudoautosomal region, compared to the remainder of the X chromosome (Additional file 2: Fig. S10 and Fig. S11). In order to determine whether the density of transposable elements within the pseudoautosomal region was greater than what was observed in other recombining regions of the genome, we randomly selected 2.5 Mb windows (the size of the pseudoautosomal region) from the autosomes to generate a null distribution of transposable element density (measured as the proportion of nucleotides occupied by a transposable element in each window). Although transposable elements have accumulated in the pseudoautosomal region, the density we observed is not outside of what is observed across the autosomes (10,000 permutations; $P = 0.135$).

## Stratum one contains a candidate sex determination gene

The master sex determination gene has not been identified in the threespine stickleback. Although master sex determination genes can be highly variable among species [2, 56], many species of fish share some common genes that have been co-opted into this role during the independent evolution of Y chromosomes. For instance, orthologs of both the anti-Müllerian hormone (*Amhy*) [57–59] as well as the anti-Müllerian hormone receptor (*Amhr2)* [60] have been used as the master sex determination gene. We searched for evidence of these genes among the annotated transcripts on the Y chromosome. We found the complete coding sequence of anti-Müllerian hormone on the Y chromosome (hereafter referred to as *Amhy*), located within the oldest stratum adjacent to the pseudoautosomal region boundary (positions 817,433–821,230). We did not locate a gametolog on the X chromosome, suggesting *Amhy* is an ancient duplication and translocation from autosome eight. Synonymous divergence between *Amhy* and its autosomal paralog exhibited synonymous divergence in range of other genes within stratum one, supporting the hypothesis that a translocation was coincident with the origin of stratum one ($d_s$ of *Amh/Amhy*, 0.423; $d_S$ interquartile range of stratum one, 0.081–0.611; $d_S$ interquartile range of stratum two, 0.027–0.075; $d_S$ interquartile range of stratum three, 0.026–0.052).

We explored whether *Amhy* had divergence patterns and expression patterns consistent with a functional role in sex determination. We aligned the protein coding sequence of AMHY to the threespine stickleback AMH paralog on autosome eight as well as to other vertebrate AMH proteins. We observed conservation of amino acids in the AMH and TGF-β domains of the protein sequence on the Y chromosome paralog that are conserved across vertebrates (Additional file 2: Fig. S12), suggesting the Y chromosome paralog is under selection in these regions to preserve function. We surveyed expression patterns of *Amhy* across the six tissues used in the gene annotations, including a larval tissue collected around the time sex determination is believed to

occur (stages 22–26 [61, 62]). *Amhy* expression was significantly higher in larval tissue compared to that in the brain (Log$_2$ fold change, – 2.031; FDR = 0.012), but expression was statistically indistinguishable when compared to testis (Log$_2$ fold change, – 0.284; FDR = 0.918) or liver (Log$_2$ fold change, – 2.054; FDR = 0.052). Additional functional genetics work is currently underway to test if this gene is necessary and sufficient for initiating male development.

## Discussion

### Evolution of the threespine stickleback Y chromosome

Using a combination of long-read sequencing and chromosome conformation capture (Hi-C) sequencing for scaffolding, we were able to assemble a highly accurate Y chromosome reference assembly for the threespine stickleback, concordant with sequenced BAC inserts and known cytogenetic markers [32]. Our new reference assembly revealed several patterns of sequence evolution that were not accurately resolved using short-read sequencing [18]. First, synonymous divergence was underestimated throughout the Y chromosome by relying on single-nucleotide polymorphisms ascertained through short-read sequencing. This effect was greatest in the oldest region of the Y chromosome (stratum one). Median $d_S$ was approximately 8.7-fold greater within stratum one when long-read sequences were used. Synonymous divergence was approximately 2.8-fold greater across the younger strata in the new reference assembly compared to the $d_S$ estimates from short-read sequencing. The short-read sequencing was also unable to distinguish two independent strata within this region, likely from a bias against aligning reads in divergent regions, leading to an under estimation of the true number of SNPs. Our results argue for caution in using short-read sequencing technologies to characterize sex-specific regions of Y or W chromosomes.

With the presence of both an X and Y chromosome reference, we were able to show that this mapping bias is alleviated, and short-read sequences can be correctly partitioned between the two chromosomes in males and females. When we analyzed nucleotide divergence between the reference Y chromosome and the short-read sequenced Y chromosomes from various populations, we found divergence was an order of magnitude lower than what was observed on the autosomes or X chromosome. Thus, threespine stickleback fish also exhibit reduced Y chromosome diversity as observed in other species [63–69]. However, there is some evidence for population divergence on the Y chromosome, as read depth was slightly lower when mapping reads from males of different populations to the Y chromosome assembly than when reads from a male of the same population were used. Additional work will be necessary to understand whether patterns of Y chromosome diversity are consistent with neutral expectations or whether nucleotide diversity is being reduced through strong selection on linked sites [63–65].

Divergence times for each of the strata can be approximated based on divergence rates between the threespine stickleback fish and the ninespine stickleback fish (*Pungitius pungitius*), which last shared a common ancestor as many as 26 million years ago [28, 30, 31]. Combined with a mean genome-wide estimate of synonymous divergence between the two species (0.184 [70]), we determined stratum one likely arose less than 21.9 million years (i.e., generations) ago, close to when the two species diverged. Using

the same calibration, stratum two formed less than 5.9 million years ago and stratum three formed less than 4.7 million years ago.

### Y chromosome centromere evolution

Due to their highly repetitive nature, centromeric arrays have been challenging to sequence and assemble using traditional approaches. However, long-read technologies have shown recent promise in traversing these inaccessible regions [14, 71, 72]. Using long-read sequencing, we were also able to recover two contigs in our assembly that contained arrays of an alpha satellite monomeric repeat that had sequence similarity to a monomeric repeat isolated from the remainder of the genome [34]. Centromeres across species are highly variable both at the level of the individual monomer and how monomers are organized at a higher level [37, 38, 73–76]. This incredible variability can even occur within species. For example, in humans, centromeric HORs are not identical between nonhomologous chromosomes [77, 78], and the Y chromosomes of mouse and humans contain divergent or novel centromeric repeats relative to the autosomes [79–81]. Consistent with these patterns, we observed a decrease in sequence similarity between the Y chromosome monomeric repeat and the consensus repeat identified from the remainder of the threespine stickleback genome [34]. We found the Y chromosome was also ordered into a complex HOR; however, we cannot determine if the structure of the Y chromosome HOR is similar or dissimilar from other threespine stickleback chromosomes. The centromere sequence from other chromosomes is currently limited to short tracts of monomeric repeats [34].

Cytogenetic work has shown the threespine stickleback Y chromosome centromere may contain a divergent satellite repeat relative to the X chromosome and autosomes [34, 82]. This hypothesis was based on a weak fluorescent in situ hybridization signal on the Y chromosome from DNA probes designed from the consensus repeat. Our Y chromosome assembly indicates a mechanism driving this pattern may be the reduced sequence identity shared between the Y chromosome monomeric repeat and the consensus monomeric repeat. An alternative explanation is that the weak hybridization signal is not due to the differences in monomeric repeat sequence, but it is actually caused by a reduction in overall size of the Y chromosome centromere. Although we isolated ~ 87 kb of centromere sequence, we did not identify a contig that spans the complete centromere, leaving the actual size of the centromere unknown. Additional sequencing work is necessary to test this alternative model.

### The genetic architecture of the threespine stickleback Y chromosome is rapidly evolving

The threespine stickleback Y chromosome is at an intermediate stage of degeneration, with the retention of a total of 44.1% of the genes present on the X chromosome, compared to the highly degenerate Y chromosomes of mammals in which only ~ 1−5% of ancestral X-linked genes remain [10, 11]. The rate of gene loss on the oldest stratum of the threespine stickleback Y, in which 82% of genes have been lost, is approximately 3.7% per million generations. This is similar to the rate of gene loss per million generations estimated for other heteromorphic sex chromosomes with similarly aged strata, such as *Rumex hastatulus* (1.1–2%) [83], *Silene latifolia* (4–8%) [17, 84], and the *Drosophila miranda* neo-Y (1.7–3.4%) [1, 84, 85]. A somewhat higher rate of gene loss

(8.4–11.5%) is found on the *Rumex rothschildianus* Y [83], but none of these systems have experienced rates of gene loss as rapid as on the similarly aged strata 4 and 5 of the primate Y chromosome (60% per million generations) [84, 86], possibly due to a lower effective population size in primates. The consistent estimates of rate of gene loss in the other plant and animal systems suggest that haploid selection in pollen is unlikely to play a major role in rates of degeneration in the plant systems examined so far (*Rumex* and *Silene*), although there is evidence that haploid-expressed genes are maintained on plant Y chromosome, just as dosage-sensitive genes are retained on animal Y chromosomes, including the threespine stickleback [10, 13, 18, 19, 83, 84, 87].

In addition to this extensive gene loss, we found acquisition of novel genes throughout all strata of the threespine stickleback Y chromosome. Although we did not detect massive amplification of gene families as observed on mammalian sex chromosomes [7, 8, 11, 20–22], many genes that had translocated from the autosomes or were present in the common ancestor of the sex chromosomes had multiple copies on the Y chromosome. The copy numbers we observed are on the same order as the duplicated genes on the sex chromosomes of multiple species of *Drosophila* [23, 49]. The gene duplications on the threespine stickleback sex chromosomes may reflect selection on the early amplification of genes important for male fertility [43] or to prevent degradation by providing a repair template through gene conversion [7, 11, 49, 54, 88–96]. Alternatively, the duplications we observe on the threespine stickleback Y chromosome may simply reflect recent translocations and duplications that have yet to degenerate and pseudogenize.

Gene expression patterns of duplicated and translocated genes suggest this process is not entirely neutral. We observed strong testis-biased expression among genes that had duplicated and translocated to the Y chromosome, similar to patterns observed on other Y chromosomes [7, 8, 11, 20–22, 46, 47, 97]. Interestingly, we observed multiple ways that testis-biased genes can accumulate on the Y chromosome. For one, many genes exhibit ancestral testis-biased expression. Genes that have translocated from the autosomes to the Y chromosome had a similar degree of testis-biased expression as the ancestral autosomal paralog. The X-linked gametologs of genes that are duplicating on the Y chromosome also had testis-biased expression ancestrally. This suggests genes can be selected to be retained on the Y chromosome because of existing male-biased expression patterns. Our observations mirror translocations on the ancient human Y chromosome; the amplified *DAZ* genes arose from an autosomal paralog that was expressed in the testis [44]. Examples of autosome-derived translocations to the Y chromosome also exist in *Drosophila* and can have ancestral testis-biased functions [46]. On the other hand, we also found that autosome-derived translocated genes evolved stronger testis-biased expression in a tissue-specific context compared to ancestral expression. The variation in testis-biased expression observed among tissue comparisons suggests the acquisition of testis functions for many genes is incomplete. This makes the threespine stickleback Y chromosome a useful system to understand the regulatory changes required for genes to evolve novel functions in the testis.

Genes that translocate to the Y chromosome either arise through RNA-mediated mechanisms or through DNA-based translocations (reviewed in [52]). Of the translocations we observed, we only detected DNA-based translocations. Work in other species has shown that DNA-based duplications occur more frequently than RNA-mediated

mechanisms [49, 52, 98, 99]. Our results support this bias on younger sex chromosomes. It is possible that the frequency of DNA-based duplications is even higher on young sex chromosomes compared to ancient sex chromosomes. DNA-based duplications are driven by erroneous double-strand break repair. On the ancient sex chromosomes of rodents, double-strand break initiation is suppressed on the sex chromosomes of males [100, 101]. This would limit the opportunity for DNA-based translocations to occur due to aberrant double-strand break repair during meiosis. However, on younger sex chromosomes, double-strand break frequencies may still be occurring at an appreciable frequency. Coupling a diverging Y chromosome with accumulating repetitive DNA would create additional opportunities for double-strand break repair through non-allelic processes, increasing the number of duplications and translocations [102].

### *Amhy* is a candidate sex determination gene

We identified the *Amhy* gene as a candidate for male sex determination in the threespine stickleback. *Amh* has been co-opted as a male sex determination gene in multiple species of fish [57–59]. The master sex determination gene is one of the primary genes that initiate evolution of a proto-Y chromosome (reviewed in [1]). Consistent with this, *Amhy* is located in the oldest region of the stickleback Y chromosome (stratum one), adjacent to the pseudoautosomal region, and synonymous divergence with its paralog is within the range of other genes in the oldest stratum. *Amhy* is expressed in developing stickleback larvae, consistent with a role in early sex determination. Finally, the amino acids that are highly conserved across vertebrates in the functional domains of the protein are also conserved on the Y chromosome paralog in stickleback fish, suggesting *Amhy* is functional. Based on the known role of AMH signaling in sex determination in other fish, and the location, expression, and sequence of the Y chromosome paralog in stickleback fish, we propose that *Amhy* is the threespine stickleback master sex determination gene. Additional functional genetics work is underway to test this hypothesis.

### Conclusions

Our threespine stickleback Y chromosome assembly highlights the feasibility of combining PacBio long-read sequencing with Hi-C chromatin conformation scaffolding to generate a high-quality reference Y chromosome assembly. With the reduction in per base pair cost associated with the newest generation of sequencers, the comparative genomics of sex chromosomes will be more accessible. This will be especially useful for taxa like stickleback fish that have multiple independently derived sex chromosome systems among closely related species [30]. This provides a unique opportunity to understand the convergent evolution of sex chromosome structure as well as the diversity of sex determination mechanisms.

### Materials and methods

#### DNA isolation and PacBio sequencing

Total DNA was isolated from a single adult male threespine stickleback that was the laboratory-reared offspring of wild-caught fish collected from the Paxton Lake benthic population (Texada Island, British Columbia). Nucleated erythrocytes were isolated from blood (extracted by repeated pipetting in bisected tissue with 0.85x SSC buffer).

High molecular weight DNA was isolated by centrifuging blood for 2 min at 2000×g, followed by resuspension of cells in 5 ml of 0.85x SSC and 27 μl of 20 μg/ml proteinase K. Cells were lysed by adding 5 mL of 2x SDS buffer (80 mM EDTA, 100 mM Tris pH 8.0, 1% SDS), followed by incubation at 55 °C for 2 min. DNA was isolated from the lysate by adding 10 mL of buffered phenol/chloroform/isoamyl-alcohol, rotating slowly at room temperature for 30 min, followed by centrifuging at 4 °C for 1 min at 2000×g. Two further extractions were performed by adding 10 mL of chloroform, rotating slowly at room temperature for 1 h, followed by centrifuging at 4 °C for 1 min at 2000×g. DNA was precipitated using 1 mL of 3 M sodium-acetate (pH 6.0) and 10 mL of cold 100% ethanol. The pellet was washed with cold 70% ethanol and resuspended in 100 μl of 10 mM Tris (pH 8.0). DNA quality was assessed on a FEMTO Pulse (Agilent, Santa Clara, CA, USA); the peak size was 132,945 bp. Size selection, library preparation, and sequencing on a PacBio Sequel platform were conducted at the Next Generation Sequencing Platform at the University of Bern (Bern, Switzerland). 37.69 Gb was sequenced across seven SMRT cells, resulting in approximately 75.25x coverage across the genome.

### PacBio assembly

Canu (v 1.7.1) [103] was used to error correct, trim, and assemble the raw PacBio reads into contigs. Default parameters were used except corOutCoverage was increased to 50 (from the default of 40) to target a larger number of reads for assembly of the sex chromosomes (the X and Y chromosomes in males have only half the available read coverage, relative to the autosomes). Increasing corOutCoverage did not substantially decrease the N50 read size for the assembly (default 40x coverage N50, 31,494 bp; 50x coverage N50, 22,133 bp). The Canu assembly was polished using Arrow (v. 2.2.2). Raw PacBio reads were first aligned to the assembled Canu contigs using pbalign (v. 0.3.1) with default parameters. Arrow was run on the subsequent alignment also using default parameters. We identified redundancy between haplotigs of the autosomal contigs by aligning all the contigs to each other using nucmer [104] and filtering for alignments between contigs that were at least 1 kb in length and had at least 98% sequence identity (to account for the elevated heterozygosity).

### Hi-C proximity guided scaffolding

The X and Y chromosomes of threespine stickleback share a considerable amount of sequence homology [18]. Hi-C proximity guided scaffolding could not accurately scaffold the X and Y chromosomes from a combined set of contigs. To simplify the scaffolding process, we separated putative X- and Y-linked contigs from the genome-wide set of contigs. Putative Y-Iinked contigs were identified as (1) contigs that aligned to the reference X chromosome, but with higher sequence divergence, and (2) contigs that only partially aligned or did not align at all to the revised female reference genome [27]. We aligned the contigs to the reference genome using nucmer in the MUMmer package (v. 4.0) [104]. Putative X- and Y-linked contigs were separated by overall sequence identity. Putative X-linked contigs were defined as having more than 25% of the contig length aligned to the reference X chromosome with a sequence identity greater than 96%, whereas putative Y-linked contigs were defined as having a sequence identity

with the reference X chromosome of less than 96%. Contigs which had less than 25% of the length aligning to the reference genome or did not align at all were retained as putative Y-linked unique sequence. Selection of the sequence identity threshold was guided by our overall ability to re-assemble the known X chromosome sequence from the set of putative X-linked PacBio Canu contigs. We tested thresholds from 92% sequence identity to 98% sequence identity and chose the threshold that resulted in the smallest size difference between the PacBio assembly and the X chromosome sequence from the reference assembly [27] (Additional file 1: Table S4). We used custom Perl scripts to separate the X- and Y-linked contigs.

Hi-C sequencing was previously conducted from a lab-reared adult male also from the Paxton Lake benthic population (Texada Island, British Columbia) (NCBI SRA database: PRJNA336561) [27]. Hi-C reads were aligned to the complete set of contigs from the Canu assembly using Juicer (v. 1.5.6) [105]. 3D-DNA (v. 180114) was used to scaffold the putative X- and Y-linked contigs separately [33, 35]. Default parameters were used except for --editor-repeat-coverage, which controls the threshold for repeat coverage during the misjoin detector step. Because Y chromosomes often have more repetitive sequence than the remainder of the genome, we scaffolded the X- and Y-linked contigs using --editor-repeat-coverage thresholds that ranged from 8 to 18. We chose the minimum threshold that resulted in a Y chromosome scaffold that maximized the total number of Y chromosome Sanger sequenced BACs that either aligned concordantly within contigs included in the scaffold or correctly spanned gaps between contigs in the scaffold (--editor-repeat-coverage 11; Additional file 1: Table S5) (see the "Alignment of BAC sequences and merging assemblies" section).

### BAC isolation and Sanger sequencing

Y-chromosome specific BACs were isolated from the CHORI-215 library [106], which was made from two wild-caught males from the same Paxton Lake benthic population (Texada Island, British Columbia, Canada) used for the PacBio and Hi-C sequencing. The Y-chromosome-specific BACs were identified using a variety of approaches. Initially, sequences surrounding known polymorphic markers (*Idh, Stn188, Stn194*) on linkage group 19 were used as probes to screen the CHORI-215 BAC library filters, and putative Y-specific BACs were identified by the presence of a Y-specific allele at that marker [32, 107]. In addition, all CHORI-215 BAC end sequences [108] were used in a BLAST (blastn) search of the threespine stickleback genome assembly, which was generated from a female [24]. All BACs for which neither end mapped to the genome or had ends that aligned to the X chromosome with elevated sequence divergence were considered as candidate Y-chromosome BACs. These candidate BACs were verified to be Y-specific using fluorescent in situ hybridization (FISH) on male metaphase spreads, following previously described protocols [32, 109]. The hybridizations were performed with CHORI-213 BAC 101E08 (*Idh*), which clearly distinguishes the X and Y chromosomes [32] labeled with ChromaTide Alexa Fluor 488-5-dUTP and the putative Y-specific BAC labeled with ChromaTide Alexa Fluor 568-5-dUTP (Invitrogen, Carlsbad, CA, USA). Starting with these verified Y-specific BACs, we then used the CHORI-215 BAC end sequences to iteratively perform an in silico chromosome walk. At each stage

of the walk, BACs were verified as Y-specific using FISH. In total, 102 BACs were sequenced.

BAC DNA was isolated from a single bacterial colony and purified on a Qiagen MaxiPrep column. DNA was sheared to 3–4 kb using Adaptive Focused Acoustics technology (Covaris, Woburn, MA, USA) and cloned into the plasmid vector pIK96 as previously described [110]. Universal primers and BigDye Terminator Chemistry (Applied Biosystems) were used for Sanger sequencing randomly selected plasmid subclones to a depth of 10x. The Phred/Phrap/Consed suite of programs was then used for assembling and editing the sequence [111–113]. After manual inspection of the assembled sequences, finishing was performed both by resequencing plasmid subclones and by walking on plasmid subclones or the BAC clone using custom primers. All finishing reactions were performed using dGTP BigDye Terminator Chemistry (Applied Biosystems, USA). Finished clones contain no gaps and are estimated to contain less than one error per 10,000 bp.

### Alignment of BAC sequences and merging assemblies

Sequenced BAC inserts were aligned to the scaffolded Y chromosome using nucmer (v. 4.0) [104]. A BAC was considered fully concordant with the PacBio Y chromosome scaffold if the following conditions were met: both ends of the alignment were within 1 kb of the actual end of the Sanger sequenced BAC, the full length of the alignment was within 10 kb of the actual length of the Sanger sequenced BAC, and the total alignment shared a sequence identity with the PacBio Y chromosome scaffold of at least 99%. BAC alignments were also identified that spanned gaps between contigs in the scaffold. An alignment that spanned gaps was considered valid if the following conditions were met: both ends of the alignment were within 1 kb of the actual end of the Sanger sequenced BAC, the total alignment length was not greater than the actual length of the Sanger sequenced BAC, and the total alignment shared a sequence identity with the PacBio Y chromosome scaffold of at least 99%. Finally, BACs were identified that extended from contigs into gaps within the scaffold but did not completely bridge the gaps. BACs that extended into gaps were identified if one end of the alignment was within 1 kb of the actual end of the Sanger sequenced BAC, the alignment extended completely to the end of a contig in the scaffold, and the total alignment shared a sequence identity with the PacBio Y chromosome scaffold of at least 99%. We used a custom Perl script to identify concordant BACs, BACs that spanned gaps in the scaffold, and BACs that extended into gaps within the scaffold.

Sanger sequenced BACs that spanned gaps and extended into gaps provided additional sequence that was not originally present in the PacBio scaffolded Y chromosome. We merged this additional sequence into the PacBio scaffold using a custom Perl script. If multiple Sanger sequenced BACs spanned a gap or extended into a gap, the BAC with the highest percent sequence identity was used.

### Alignment of whole-genome short-read sequencing

We verified PacBio sequencing accuracy by comparing it to aligned Illumina paired-end short-read sequencing from male and female fish from multiple populations: one male and one female from a Puget Sound population (Washington State, USA; NCBI

SRA database accessions: SRR6954368 and SRR6954353 [114]), one male and one female from a Lake Washington population (Washington State, USA; NCBI SRA database accession: SRR6954338 and SRR6954339 [114]), one female from the Paxton Lake limnetic population (Texada Island, British Columbia, Canada; NCBI SRA database accession: SRR5626528), and a different male from the same Paxton Lake benthic population used to sequence the reference Y chromosome (NCBI SRA database accession: SRR5626529). Reads were quality trimmed with Trimmomatic (v. 0.36) [115] using a sliding window of 4 bases, trimming the remainder of the read when the average quality within a window dropped below 15. Reads were only retained if they were at least 75 nucleotides long after trimming. Trimmed paired-end reads were aligned to the revised threespine stickleback reference assembly [27] and the newly assembled reference Y chromosome with Bowtie2 (v. 2.3.5.1) [116], using default parameters. PCR duplicates were identified and removed using MarkDuplicates of Picard Tools (default parameters v. 2.21.6) [117]. Only reads with a mapping quality score of 20 or greater were retained. Genome coverage was quantified at every base across the genome using the genomecov tool of bedtools (default parameters; v. 2.29.2) [118]. Median read depth was calculated in 1 kb non-overlapping windows across the genome.

Single-nucleotide polymorphisms were genotyped in the three male samples using the Genome Analysis Toolkit (GATK; v. 4.1.2), following the best practices for variant discovery [119]. Variants were called using HaplotypeCaller (-ERC GVCF), and joint genotyping was conducted using GenotypeGVCFs (default parameters). Indels and heterozygous sites were removed using bcftools (v. 1.9). Heterozygous sites were not considered when calculating pairwise sequence divergence because these genotypes likely reflect collapsed paralogs on the Y chromosome reference relative to the aligned population or may reflect incorrectly aligned X-linked reads to some regions on the Y chromosome. We filtered heterozygous sites throughout the genome.

### Identification of the Y centromere

The Y chromosome centromere was localized using chromatin immunoprecipitation targeting centromere protein A (CENP-A) as previously described [34]. The threespine stickleback-specific antibody against CENP-A is readily available from the authors. Immunoprecipitated and input DNA from two males from the Japanese Pacific Ocean population (Akkeshi, Japan) were 150-bp paired-end sequenced using an Illumina HiSeq 2500. Reads were quality trimmed with Trimmomatic (v. 0.36) [115] using a sliding window of 4 bases, trimming the remainder of the read when the average quality within a window dropped below 15. Trimmed paired-end reads were aligned to the scaffolded Y chromosome assembly with Bowtie2 (v. 2.3.4.1) [116], using default parameters. This resulted in an overall alignment rate of 83.9% (chromatin only input) and 81.0% (immunoprecipitation) for the first male and 82.3% (chromatin only input) and 79.8% (immunoprecipitation) for the second male. We quantified the read depth of aligned reads at every position across the Y chromosome using the genomecov package of bedtools (v. 2.28.0) [118]. We calculated fold-enrichment of reads in the immunoprecipitation versus the input DNA at every position across the Y chromosome. Each position was normalized by the total number of reads in the respective sample before calculating the immunoprecipitation to input DNA ratio. The mean fold-enrichment

was calculated every 1 kb across the Y chromosome. Fold-enrichment was quantified using a custom Perl script.

The autosomal core centromere repeat (GenBank accession KT321856) [34] was aligned to the Y centromere region using BLAST (blastn) [120]. Only hits that had an alignment length ± 10 bp of the core 187 bp repeat were retained. Average percent identity was calculated among the remaining BLAST hits. We determined a majority consensus sequence from the core 14 centromere repeat units from the initial Y chromosome assembly. The majority consensus was used to identify additional repeats in the "debris" fragments that flanked the gap in the scaffold where the Y centromere was originally identified. The majority consensus was aligned to the debris fragments using BLAST (blastn), retaining alignments that had an alignment length ± 10 bp of the core 187 bp repeat. Pairwise alignments between all repeats within the Y chromosome were conducted using BLAST (blastn). Average percent identity among all pairwise alignments was calculated using a custom Perl script.

### Molecular evolution of genes on the Y chromosome

To characterize divergence between ancestral genes shared by the X and Y chromosomes, we aligned the coding sequence of each Ensembl predicted gene to the Y chromosome using Exonerate (v. 2.4.0) [121] using the parameters --model est2genome --bestn 15. Only coding sequences for which at least 95% of its sequence length aligned to the Y chromosome were retained for further analysis. $d_S$ and $d_N$ were quantified for each pairwise alignment using the codeml module of PAML (phylogenetic analysis by maximum likelihood) (runmode = 2) [122]. If an X coding sequence aligned to multiple locations on the Y chromosome, only the alignment with the lowest $d_S$ was retained. In addition, all alignments with $d_S$ greater than two were removed. These stringent filtering steps aimed to limit alignments to the true gametolog, rather than to paralogs of genes that are present in greater than one copy on the sex chromosomes. For estimating $d_N/d_S$, transcripts with a value of 99 were omitted. Strata breakpoints were broadly based upon the inversion breakpoints in the cytogenetic map [32], adjusted at a fine-scale by the inversion breakpoints in the alignments between the assembled Y chromosome and the reference X chromosome (breakpoints on the Y chromosome: PAR/stratum one, 0.34 Mb; stratum one/stratum two, 4.67 Mb; stratum two/stratum three, 9.67 Mb; breakpoints on the X chromosome: PAR/stratum two, 2.5 Mb; stratum two/stratum three, 6.89 Mb; stratum three/stratum one, 12.5 Mb). The pseudoautosomal region boundary was set at 2.5 Mb on the X chromosome, as previously defined by patterns of molecular divergence between the X and Y chromosome [18] and through genetic linkage maps [25, 107, 123].

### Identification of haploinsufficient genes

One-to-one human-threespine stickleback fish orthologs were identified from the Ensembl species comparison database. Orthologs were restricted to those with a human orthology confidence of 1. The high confidence orthologs were matched to the human haploinsufficiency predictions from the DECIPHER database (v. 3) [41, 42].

## Gene annotation across the PacBio assembled Y chromosome

Genes were annotated on the repeat masked Y chromosome scaffold using the MAKER genome annotation pipeline (v. 3.01.02) [50, 51] using evidence from multiple RNA-seq transcriptomes, all predicted protein sequences from Ensembl (release 95), and ab initio gene predictions from SNAP [124] and Augustus [125]. RNA-seq was conducted on multiple tissue samples. RNA from adult male whole brains was previously extracted and sequenced from wild-caught fish from the Japanese Pacific Ocean population, Akkeshi, Japan (NCBI BioProject accession: PRJNA277770) [18]. Male larval tissue was collected from stages 22–26 [61] when sex determination is believed to occur [62]. Larvae were collected from laboratory-reared progeny of wild-caught fish from the Lake Washington population (Seattle, Washington). Larvae were pooled into two samples, one consisting of five males and the other consisting of six males. Total RNA was extracted using TRIzol reagent (Invitrogen, USA) following standard protocols. Library preparation and sequencing was conducted by the Fred Hutchinson Cancer Research Center Genomics Shared Resource. Single-end sequencing was carried out on a Genome Analyzer II for 72 cycles. Liver and testis tissues were also collected from adult and juvenile fish from laboratory-reared progeny of wild-caught fish from the Japanese Pacific Ocean population (Akkeshi, Japan). Livers and testes were collected from two males and pooled. Three juvenile samples and three adult samples were collected. Total RNA was extracted using TRIzol reagent (Invitrogen, USA) following standard protocols. Library preparation and sequencing was conducted by the Georgia Genomics and Bioinformatics Core at the University of Georgia. Paired-end sequencing was carried out on a NextSeq 500 for 150 cycles. All reads were quality trimmed with Trimmomatic (v. 0.36) [115] using a sliding window of 4 bases, trimming the remainder of the read when the average quality within a window dropped below 15.

We aligned sequences to the masked revised whole-genome reference assembly [27] using Tophat (v. 2.3.4.1) [126]. Default parameters were used except for the liver and testis tissues. For these tissues, we used --read-mismatches 4 and --read-edit-dist 4 to account for the greater number of SNPs in the 150 bp reads. These alignment parameters produced an overall alignment rate to the masked genome of 80.4% for the brain tissue, 68.0% in the adult liver tissue, 64.5% in the juvenile liver tissue, 64.7% for adult testis tissue, 65.5% for the juvenile testis tissue, and 68.9% for the larval tissue. Aligned reads from all samples within a tissue were pooled to construct a single tissue-specific set of transcripts using Cufflinks (v. 2.2.1) [127] with default parameters. Exons from the GTF file were converted to FASTA sequences with gffread.

MAKER was run over three rounds. For the first round of MAKER, we only used evidence from the RNA-seq transcripts and all annotated protein sequences from Ensembl (release 95) using default parameters and est2genome=1, protein2genome=1 to infer gene predictions directly from the transcripts and protein sequences. We used these gene models to train SNAP. In addition, Augustus was trained using gene models from BUSCO conserved orthologs found on the PacBio scaffolded Y chromosome and the revised reference assembly [26, 27] with the Actinopterygii dataset and default BUSCO (v. 3.0.2) parameters [128, 129]. MAKER was run using the new SNAP and Augustus models with est2genome=0 and protein2genome=0. For the third round of MAKER, SNAP was retrained with the updated gene models and MAKER was run again with the updated SNAP model, the previous Augustus model, and est2genome=0 and

protein2genome=0. The threespine stickleback repeat library derived using RepeatModeler was used during the annotation pipeline using the rmlib option.

To characterize whether there were any novel genes acquired by the Y chromosome as well as any duplicated genes, we aligned each MAKER annotated gene on the PacBio assembled Y chromosome to the whole genome as well as back to the Y chromosome, following the same exonerate procedure. If a paralog was identified on an autosome, we only retained the paralog if $d_S$ was lower than the median $d_S$ across the oldest region of the Y chromosome (stratum one: 0.101). Using this stringent filter avoids incorrectly assigning more ancient paralogs on the autosomes as the most recent common ancestor. If multiple alignments were identified on the X chromosome, only the alignment with the lowest $d_S$ was retained. If multiple overlapping paralogs from a single gene were identified on the Y chromosome, only the paralog with the lowest $d_S$ was retained. Alignments to the unassigned contigs (ChrUn) were ignored because these contigs could not be unambiguously assigned to the X chromosome or to the autosomes.

### Differential expression of genes on the Y chromosome

For each tissue used in the gene annotations, the total number of RNA reads that mapped to the reference Y chromosome were counted using htseq-count (HTSeq software package; v. 0.9.1) [130]. Read counts were obtained across all 626 MAKER identified genes across the male-specific region of the Y chromosome plus all additional paralogs (132 paralogs). Default parameters were used with the addition of --stranded=no and --nonunique all. Ambiguous reads were included in the counts because of the large number of paralogs on the Y chromosome with high sequence identity. In case a read could not be unambiguously mapped, it was assigned to all features to which it matched. Genes were removed from the analysis if they had a read count of zero in all samples. Scaling factors for normalization were calculated using the trimmed mean of $M$-values (TMM) method in the Bioconductor package, edgeR [131]. The TMM method minimizes the log-fold changes between samples for most genes. This approach may not be appropriate for a Y chromosome, which may be enriched for male-biased gene expression. Therefore, we calculated scaling factors for all autosomal transcripts and normalized the Y chromosome transcripts using these scaling factors. Ensembl annotated transcripts were used for the autosomes. Replicates were grouped based on tissue (testis: six samples; liver: six samples; brain: three samples; larvae: two samples). Log$_2$ fold-change was calculated for each gene in each tissue comparison using edgeR.

### Repetitive element annotation

Repetitive elements were first modeled together on the PacBio scaffolded X and Y chromosomes using RepeatModeler (v. 1.0.11) [132] with default parameters. Repeats were masked across both scaffolds using RepeatMasker (v. 4.0.7) [133] with default parameters and the custom database created by RepeatModeler.

### Characterization of *Amhy*

The protein sequence of AMHY was aligned to AMH sequences from human (GenBank AAH49194.1), mouse (GenBank NP_031471.2), chicken (GenBank NP_

990361.1), zebrafish (GenBank AAX81416.1), and the paralog of AMH on threespine stickleback chromosome eight (ENSGACP00000016697) using CLUSTALW with default parameters in Geneious Prime (v. 2019.1.1) [134]. Synonymous divergence was estimated between *Amhy* and the paralog on autosome eight (ENSGACT00000016731.1) using the codeml module of PAML (runmode = 2) [122]. Gene expression level was quantified for *Amhy* in the six different tissues used for gene annotation. Read counts per million (CPM) for each tissue was calculated from the TMM-scaled samples from the differential expression analysis.

## Supplementary information

---

**Additional file 1.** Supplementary tables.

**Additional file 2.** Supplementary figures.

**Additional file 3.** Review history.

---

### Acknowledgements
The authors acknowledge Ivan Liachko and Shawn Sullivan (Phase Genomics) for assistance with Hi-C scaffolding, and the following core facilities for sequencing: Fred Hutchinson Cancer Research Center Genomics Shared Resource, Next Generation Sequencing Platform of the University of Bern, and The Georgia Genomics and Bioinformatics Core at the University of Georgia. Arne Nolte, Joana Meier, and Marcel Häsler provided advice on the isolation of high molecular weight DNA. Amanda Bruner provided guidance in staging the threespine stickleback embryos.

### Peer review information

### Review history
The review history is available as Additional file 3.

### Authors' contributions
CLP conceived the experiments, acquired and interpreted data, and revised the manuscript. SRM acquired and interpreted data and revised the manuscript. JAR acquired and interpreted data and revised the manuscript. AFSN acquired and interpreted data and revised the manuscript. JRU acquired and interpreted data and revised the manuscript. JNC acquired and interpreted data and revised the manuscript. JG acquired and interpreted data. JS acquired and interpreted data. RMM acquired and interpreted data. DMK acquired and interpreted data and revised the manuscript. MAW conceived the experiments, acquired and interpreted data, and drafted and revised the manuscript. All authors read and approved the final manuscript.

### Funding
The authors acknowledge the following funding sources: NIH R01 GM071854 (CLP), NIH P50 HG002568 (RMM, DMK, CLP), NIH R01 GM116853 (CLP), Fred Hutchinson Cancer Research Center (CLP), Swiss National Science Foundation 31003A_176130 (CLP), Evolutionary, Ecological, or Conservation Genomics Research Award from the American Genetic Association (MAW), Howard Hughes Medical Institute and the Life Sciences Research Foundation (MAW), NIH Cell and Molecular Biology Training Grant T32 GM07270 (JAR), NIH Genome Training Grant T32 HG00035 (JRU), NIH Chromosome Metabolism and Cancer Training Grant T32 CA009657 (JNC), National Science Foundation Graduate Research Fellowship DGE 1256082 (JNC), Office of the Vice President of Research at the University of Georgia (MAW), National Science Foundation IOS 1645170 (MAW), and National Science Foundation MCB 1943283 (MAW).

### Availability of data and materials
The datasets supporting the conclusions of this article are included within the article, within Additional file 1 (supplemental tables), Additional file 2 (supplemental figures), and within the Short Reads Archive database (https://www.ncbi.nlm.nih.gov/sra) under accession numbers PRJNA591630 [135], PRJNA336561 [27], and PRJNA277770 [18], SRR6954368 [114], SRR6954353 [114], SRR6954338 [114], SRR6954339 [114], SRR5626528 [136], and SRR5626529 [137]. Custom Perl scripts are available at https://github.com/mikewhitelab/Y_Chromosome_Assembly [138]. The Y chromosome assembly is also available for download from the threespine stickleback genome browser at https://stickleback.genetics.uga.edu [139].

### Ethics approval and consent to participate
All procedures using animals were approved by the Fred Hutchinson Cancer Research Center Institutional Animal Care and Use Committee (protocol 1575), the Veterinary Service of the Department of Agriculture and Nature of the Canton of Bern (protocol BE17/17), and the University of Georgia Animal Care and Use Committee (protocol A2018 10-003-R1).

### Consent for publication
Not applicable.

**Competing interests**
The authors declare that they have no competing interests.

**Author details**
[1]Divisions of Human Biology and Basic Sciences, Fred Hutchinson Cancer Research Center, Seattle, WA 98109, USA. [2]Institute of Ecology and Evolution, University of Bern, Baltzerstrasse 6, 3012 Bern, Switzerland. [3]Graduate Program in Molecular and Cellular Biology, University of Washington, Seattle, WA 98195, USA. [4]Department of Genetics, University of Georgia, Athens, GA 30602, USA. [5]HudsonAlpha Institute for Biotechnology, Huntsville, AL 35806, USA. [6]Department of Developmental Biology and Howard Hughes Medical Institute, Stanford University School of Medicine, Stanford, CA 94305, USA.

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

## 