## [**Additional file 3.** Review history. · Genome Biology]

Review History

First round of review

Reviewer 1

Are you able to assess all statistics in the manuscript, including the appropriateness of statistical tests used?

Yes, and I have assessed the statistics in my report.

Comments to author:

This manuscript reports the sequencing, assembly, and substantive analysis of a young vertebrate Y chromosome. The authors used a combination of sequencing strategies, blending long-read PacBio, HiC, and targeted BAC sequencing via Sanger to create a high-quality and very contiguous Y chromosome assembly. They subsequently analyzed patterns of architectural rearrangements and divergence strata, gene content evolution as well as transposon abundance, and centromeric repeats. Finally, they identified a likely candidate male-determining loci, notably in the oldest stratum, consistent with predictions from theory concerning Y chromosome evolution.

Overall I think this is a very strong manuscript that is very well written and organized, communicating well-executed research that is informatively contextualized and interpreted. I do not find any major flaws or points to critique. I have only a couple of minor comments and suggestions that hopefully serve to marginally improve the manuscript:

Are there flow-cytometry estimates of genome size specific to male and female? If so, you could calculate the expected size difference between X and Y... ($X - Y = 2C_{\text{female}} - 2C_{\text{male}}$). Since you have a good X assembly, you could then estimate the size of the Y independently of your assembly to evaluate how complete it is. Probably it is quite complete, but this could be one nice detail to add, assuming you had such estimates on hand. If you don't have them, then never mind.

Fig2 I think it would be better to include a color legend indicating strata, rather than writing it in the legend. A legend would probably fit nicely in the lower left. Perhaps also in SuppFig 7.

L331 "Within strata one and two, we found orthologs with a retained Y-linked allele had lower DHP scores than genes without a Y ortholog..." I think there is an inconsistent, slightly confusing, and potentially erroneous use of the terms "ortholog" and "allele" here. First, it is confusing because the first use of "orthologs" refers (I think) to X-linked stickleback genes that have human orthologs with DHP scores. You then (I think) mean to contrast these stickleback-X genes as having (or not having) Y-linked homologs, which you alternatively refer to as orthologs (colliding with the previous use of this term) and then alleles. However, I think the most correct term here would be "gametolog" to refer to the Y-linked homologs.

L376 Similar to above, "X-linked allele" might be better expressed as a gametolog.

L389-395 While I agree that the patterns presented in SuppFig 7 do generally point to a higher density, I think it would be worthwhile to quantify this a bit more precisely to support this claim. Could you do a window-based analysis to generate a distribution of the per-base content of TEs on each section of the Y relative to X, or between Y regions?

Reviewer 2

Are you able to assess all statistics in the manuscript, including the appropriateness of statistical tests used?

Yes, and I have assessed the statistics in my report.

Comments to author:

This ms by Peichel et al. reports the assembly of a largely non-recombining Y chromosome in fish. They used a combination of PacBio sequencing and Hi-C to obtain a high-quality assembly of a ~16 Mb Y chromosome in stickleback. Analyzing the Y sequence, they found its centromere, three evolutionary strata, an enrichment of haploinsufficient genes and an accumulation of testis-biased expression genes on the Y. Contrary to mammalian Y chromosomes, they did not find any palindrome or ampliconic structure. They also proposed a candidate sex-determining gene (*Amhy*) based on work in other fish.

So far, largely non-recombining Y chromosomes have been sequenced in mammals only. The stickleback Y chromosome is thus a significant advance for the study of vertebrate sex chromosomes. The analyses presented in this ms are interesting and consistent with previous work on fragmented data on stickleback sex chromosomes. I think the most striking findings are 1) the finding of a candidate master sex-determining gene 2) the absence of palindromes or other ampliconic structures found in mammals. On this ground, I think that this ms merits a publication in *Genome Biology*, pending some modifications listed below.

1) They need to be clearer when presenting the literature and key definitions about sex chromosomes. Otherwise, many statements are too vague and prevent the reader to really understand the advance provided by this work. Some important references are missing. Also, a more precise definition of key terms will make the Discussion part clearer. In the abstract and in the introduction, the author claimed that "Y chromosome sequence assemblies have only been generated across a handful of taxa with ancient sex chromosomes." and "Reference assemblies of young sex chromosomes are largely absent". Only papers on mammals and *Drosophila* are cited.

I think the authors need to make a distinction between heteromorphic vs. homomorphic sex chromosomes or sex chromosomes with small vs. large non-recombining regions. They also need to be more precise than "old" or "young" to characterize sex chromosomes.

It is correct that only few Y chromosomes have been sequenced and assembled compared with the number of sequenced and assembled genomes. However, there are more of them than the mammalian and *Drosophila* ones. Several homomorphic Y chromosomes have been sequenced and assembled (e.g. papaya, Wang et al. 2012; medaka, Kondo et al. 2006). The authors have only cited work on heteromorphic Y chromosomes. They should either specify it or cite work on homomorphic Y chromosomes as well. It is not to diminish their work, sequencing heteromorphic Y chromosomes is more difficult than sequencing homomorphic Y ones, but again I think it's better to be precise. Note that W, U and V chromosomes (also non-recombining as the Y) have also been sequenced but they can be neglected here...

It is not the first study using PacBio to sequence Y chromosomes. Single molecule sequencing has a strong potential for getting high-quality assemblies of the Y chromosomes because of the long reads it generates. They cite a paper on the *Drosophila miranda* neoY, but PacBio has been used successfully in other species (e.g. gorilla, Tomaszewicz et al. 2016; mosquitoes, catfish; Bao et al. 2019). This should be acknowledged.

Some of the homomorphic Y chromosomes that were sequenced are a few millions years old. The statement that no young Y chromosomes has been sequence before is not correct. Throughout the ms, the XY stickleback system is said to be "young" because it is less than 26 MY old. But the stickleback XY system is clearly heteromorphic. Young / old Y chromosomes: what does it mean?

Several authors have warned against using absolute age to compare sex chromosome systems (Bachtrog 2011; Muyle et al. 2017). Species can have drastically different generation time and 1 MY for a mammal is not the same as 1 MY for a fly. To make fair comparison, one has to correct for the generation time. Another possibility is to use raw synonymous divergence instead of absolute times obtained from molecular clock.

In stickleback, the X-Y dS of the oldest stratum is 0.15 (see Figure 4 and Table 1 of this ms). It is similar to that of *Silene latifolia* (a plant XY heteromorphic system, also with several strata, e.g. Papadopoulos et al. 2015) and is higher than the X-Y dS of the strata 4 and 5 in primates. It is not that young and should be distinguished from incipient systems as the medaka XY one, for example.

Referring to the steps of the evolution of heteromorphic sex chromosomes would be appropriate here. The

stickleback system does not fit in with the first steps, it is more alike to the intermediate steps where heteromorphy is already visible (e.g Bachtrog 2013; Charlesworth et al. 2005).

I agree this problem is not specific to this manuscript and in the past every system younger than the mammalian Y (=180 MY old) would be called young, but this is not precise and should be avoided.

In line with this, I do not agree with this statement "The young sex chromosome of threespine stickleback is a useful model system to understand how the genetic architecture of sex-limited chromosomes initially evolves." I think the stickleback XY system is a useful system that provides insight on the intermediary steps of the evolution of heteromorphy, not the very first steps. To study the first steps, incipient sex chromosomes with tiny non-recombining regions are more relevant (e.g. Charlesworth 2019).

In the Discussion: "Despite the young age of the threespine stickleback Y chromosome relative to mammals" is too vague... See my comments above to rephrase it.

2) I would welcome a bit more details on how the assembly was done. It is indicated in the ms that the individual that was sequenced was heterozygous. In this case, Falcon and Falcon unzip are known to perform really well. It would be nice to have a justification of why Canu was used here instead. Also, the way the X and Y chromosomes were assembled separately is not entirely clear. A high-quality assembly of the Y should indeed not be a chimeric assembly with bits of X and Y but this is not easy when using DNA from a male genome and not from flow sorted Y chromosome or Y-probe screened BAC libraries as in some of the previous work cited in the introduction. It would be nice to include some tests showing that the X and the Y here are not chimeric. The result of the mapping of reads from a female genome on the X and Y references could be interesting. Identifying of Y-linked SNPs from multimale - multifemale comparison and checking that these are present on the Y reference only could be too.

3) It is not clear to me how the PAR was identified here. This should be explained more precisely. Also, I would add the PAR in Figure 4.

4) I think it would be nice to include an analysis of Y degeneration here. What is the extent of gene loss on the Y chromosome of stickleback? It would be really interesting to compare that number with other comparable systems. The best option would be to use a fish with a different sex chromosome system as an outgroup and to infer gene loss (and gain) on the X and Y separately in stickleback. An easier option (assuming the gene content of the X has not changed too much) is to compare directly X and Y chromosomes' gene content in stickleback.

5) They found that TEs are abundant in the PAR. This is not expected. TEs are expected to accumulate in non-recombining parts of the sex chromosomes. PARs are typically highly recombining and are not expected to contain many repeats. This should be discussed (e.g. is recombination rate low in the stickleback PAR?). A similar pattern has been found in the brown alga *ectocarpus* (Luthringer et al. 2015).

6) Their proposition that *Amhy* is the master sex-determining gene in stickleback is interesting. Of course, genetic validation and functional studies are required to really test this idea. One thing that could be checked here is the dS value of the *AmhY* with its autosomal parent. This dS should be consistent with the mean dS of the stratum 1 (or at least fall in the range of dS values of that stratum) if the evolution of *Amhy* as a master sex-determining gene is associated with the evolution of stratum 1.

7) p 22, there is a discussion on the differences between dS values estimated from Illumina assemblies and from PacBio assemblies. dS values are found higher in the latter than in the former. It is well known that PacBio has a high sequencing error rate. One immediately wonders whether the differences in dS result from this and is a mere methodological artefact. How much the auto-correction of the PacBio contigs and the polishing were successful should probably be discussed here to rule out his possibility.

8) The proposed scenario for the formation of the 3 strata (shown in Figure S9) does not make sense to me. As far as I understand the figure, it looks like in step 2, there is a recombining region between strata 1 and 2 (in addition to the PAR at the tip of the sex chromosomes). This implies that strata 1 and 2 would not be genetically linked at this stage. I am not saying that is impossible. Actually, it would be very interesting. Theory says indeed that we should not observe such an event. In the current model for the evolution of heteromorphy, the non-recombining region on

the Y tends to grow over time by genetically linking fragments of the Y to the male-determining gene(s). It seems to me that other scenarios of the chromosomal rearrangements between X and Y chromosomes should be explored and discussed. What about an inversion on the X chromosome encompassing strata 2 and 3 following the formation of strata 3? This has happened on the bird Z chromosomes (Zhou et al. 2014; Yazdi & Ellegren 2018). Using an outgroup (with a different sex chromosome system) could help pinpoint the inversions that took place between the stickleback X and Y chromosomes.

9) The analysis on testis-biased gene accumulation is interesting but I am not sure that the approach they used is the best suited. They compared the accumulation of tissue-specific genes on the Y comparing three tissues: testis, brain and liver. Their results, however, could be explained by a genome-wide excess of testis-biased genes compared to other tissues and not a Y-specific excess. To know for sure, one would need to compare all chromosomes. The studies that have established an accumulation of sex-biased genes on the sex chromosomes have done so (reviewed in Parsch & Ellegren 2007, 2013; and see for example Assis & Bachtrog 2012). Here, I think looking at the distribution of testis-biased gene in the genome would be required to back up the claim that the Y is enriched in those genes.

10) As mentioned above, the stickleback XY and the *Silene latifolia* XY systems have similar level of heteromorphy and age ($dS \sim 0.15$, 3 strata, significant differences in X/Y size). It is noteworthy that the stickleback Y chromosome is smaller than the X, when the *S. latifolia* Y is larger than the X (also observed in *Coccinia grandis*, another plant heteromorphic system similar to *S. latifolia*, Sousa et al. 2013). This raises the possibility that the evolution of Y size may be different in animals and plants, an interesting idea.

References

- Assis R, Zhou Q, Bachtrog D. Sex-biased transcriptome evolution in *Drosophila*. *Genome Biol Evol.* 2012;4(11):1189-1200. doi:10.1093/gbe/evs093
- Bachtrog D. Plant sex chromosomes: a non-degenerated Y?. *Curr Biol.* 2011;21(18):R685-R688. doi:10.1016/j.cub.2011.08.027
- Bachtrog D. Y-chromosome evolution: emerging insights into processes of Y-chromosome degeneration. *Nat Rev Genet.* 2013;14(2):113-124. doi:10.1038/nrg3366
- Bao L, Tian C, Liu S, et al. The Y chromosome sequence of the channel catfish suggests novel sex determination mechanisms in teleost fish. *BMC Biol.* 2019;17(1):6. Published 2019 Jan 25. doi:10.1186/s12915-019-0627-7
- Charlesworth D, Charlesworth B, Marais G. Steps in the evolution of heteromorphic sex chromosomes. *Heredity (Edinb).* 2005;95(2):118-128. doi:10.1038/sj.hdy.6800697
- Charlesworth D. Young sex chromosomes in plants and animals. *New Phytol.* 2019;224(3):1095-1107. doi:10.1111/nph.16002
- Ellegren H, Parsch J. The evolution of sex-biased genes and sex-biased gene expression. *Nat Rev Genet.* 2007;8(9):689-698. doi:10.1038/nrg2167
- Hall AB, Papatianos PA, Sharma A, et al. Radical remodeling of the Y chromosome in a recent radiation of malaria mosquitoes. *Proc Natl Acad Sci U S A.* 2016;113(15):E2114-E2123. doi:10.1073/pnas.1525164113
- Kondo M, Hornung U, Nanda I, et al. Genomic organization of the sex-determining and adjacent regions of the sex chromosomes of medaka. *Genome Res.* 2006;16(7):815-826. doi:10.1101/gr.5016106
- Luthringer R, Lipinska AP, Roze D, et al. The Pseudoautosomal Regions of the U/V Sex Chromosomes of the Brown Alga *Ectocarpus* Exhibit Unusual Features. *Mol Biol Evol.* 2015;32(11):2973-2985. doi:10.1093/molbev/msv173
- Muyle A, Shearn R, Marais GA. The Evolution of Sex Chromosomes and Dosage Compensation in Plants. *Genome Biol Evol.* 2017;9(3):627-645. doi:10.1093/gbe/evw282
- Papadopulos AS, Chester M, Ridout K, Filatov DA. Rapid Y degeneration and dosage compensation in plant sex chromosomes. *Proc Natl Acad Sci U S A.* 2015;112(42):13021-13026. doi:10.1073/pnas.1508454112
- Parsch J, Ellegren H. The evolutionary causes and consequences of sex-biased gene expression. *Nat Rev Genet.* 2013;14(2):83-87. doi:10.1038/nrg3376
- Sousa A, Fuchs J, Renner SS. Molecular cytogenetics (FISH, GISH) of *Coccinia grandis*: a ca. 3 myr-old species of cucurbitaceae with the largest Y/autosome divergence in flowering plants. *Cytogenet Genome Res.* 2013;139(2):107-118. doi:10.1159/000345370
- Tomaszkiewicz M, Rangavittal S, Cechova M, et al. A time- and cost-effective strategy to sequence mammalian Y Chromosomes: an application to the de novo assembly of gorilla Y. *Genome Res.* 2016;26(4):530-540.

doi:10.1101/gr.199448.115

Wang J, Na JK, Yu Q, et al. Sequencing papaya X and Yh chromosomes reveals molecular basis of incipient sex chromosome evolution. *Proc Natl Acad Sci U S A*. 2012;109(34):13710-13715. doi:10.1073/pnas.1207833109

Yazdi HP, Ellegren H. A Genetic Map of Ostrich Z Chromosome and the Role of Inversions in Avian Sex Chromosome Evolution. *Genome Biol Evol*. 2018;10(8):2049-2060. Published 2018 Aug 1.

doi:10.1093/gbe/evy163

Zhou Q, Zhang J, Bachtrog D, et al. Complex evolutionary trajectories of sex chromosomes across bird taxa. *Science*. 2014;346(6215):1246338. doi:10.1126/science.1246338

May 4, 2020

Dear Dr. Pang,

We are pleased to submit a revised version of our manuscript, “Assembly of the threespine stickleback Y chromosome reveals convergent signatures of sex chromosome evolution.”

We believe we have addressed the previous concerns of the reviewers. In response, we have now incorporated a thorough discussion of sex chromosomes at different stages of degeneration and placed our results in the context of these systems. We have also included multiple Illumina short-read sequencing datasets that have verified the overall accuracy of the Y chromosome assembly. We hope that these changes will satisfy your concerns and those of the reviewers. Please see our point-by-point response to the reviewers below.

Sincerely,
Michael A. White on behalf of all the authors
Department of Genetics
University of Georgia
whitem@uga.edu

Reviewer #1: This manuscript reports the sequencing, assembly, and substantive analysis of a young vertebrate Y chromosome. The authors used a combination of sequencing strategies, blending long-read PacBio, HiC, and targeted BAC sequencing via Sanger to create a high-quality and very contiguous Y chromosome assembly. They subsequently analyzed patterns of architectural rearrangements and divergence strata, gene content evolution as well as transposon abundance, and centromeric repeats. Finally, they identified a likely candidate male-determining loci, notably in the oldest stratum, consistent with predictions from theory concerning Y chromosome evolution.

Overall I think this is a very strong manuscript that is very well written and organized, communicating well-executed research that is informatively contextualized and interpreted. I do not find any major flaws or points to critique. I have only a couple of minor comments and suggestions that hopefully serve to marginally improve the manuscript:

Are there flow-cytometry estimates of genome size specific to male and female? If so, you could calculate the expected size difference between X and Y... ($X - Y = 2C_{\text{female}} - 2C_{\text{male}}$). Since you have a good X assembly, you could then estimate the size of the Y independently of your assembly to evaluate how complete it is. Probably it is quite complete, but this could be one nice detail to add, assuming you had such estimates on hand. If you don't have them, then never mind.

Response: We agree that flow cytometry estimates of male and female genome size would help us determine how complete our Y chromosome assembly is. Unfortunately, however, the existing papers that used flow cytometry to estimate the sizes of a variety of fish and/or vertebrate genomes only measured one individual of unspecified sex

(Hinegardner and Rosen (1972) American Naturalist 106: 621-644; Vinogradav (1998) Cytometry 31: 100-109).

Fig2 I think it would be better to include a color legend indicating strata, rather than writing it in the legend. A legend would probably fit nicely in the lower left. Perhaps also in SuppFig 7.

Response: We thank the reviewer for this suggestion. We now include a legend in Figure 2 and Supplemental Figure 10 (formerly Supplemental Figure 7).

L331 "Within strata one and two, we found orthologs with a retained Y-linked allele had lower DHP scores than genes without a Y ortholog..." I think there is an inconsistent, slightly confusing, and potentially erroneous use of the terms "ortholog" and "allele" here. First, it is confusing because the first use of "orthologs" refers (I think) to X-linked stickleback genes that have human orthologs with DHP scores. You then (I think) mean to contrast these stickleback-X genes as having (or not having) Y-linked homologs, which you alternatively refer to as orthologs (colliding with the previous use of this term) and then alleles. However, I think the most correct term here would be "gametolog" to refer to the Y-linked homologs.

Response: We agree that the use of the term "ortholog" was confusing. We replaced all instances of "Y-linked homologs" and "alleles" with "gametolog."

L376 Similar to above, "X-linked allele" might be better expressed as a gametolog.

Response: We replaced all instances of "X-linked allele" with "gametolog."

L389-395 While I agree that the patterns presented in SuppFig 7 do generally point to a higher density, I think it would be worthwhile to quantify this a bit more precisely to support this claim. Could you do a window-based analysis to generate a distribution of the per-base content of TEs on each section of the Y relative to X, or between Y regions?

Response: We thank the reviewer for this suggestion. We now include a new supplemental figure (Supplemental Figure 11) that shows there is a higher density of transposable elements per nucleotide in each stratum of the Y chromosome, compared to each stratum of the X chromosome. This figure also highlights that the transposable element density is slightly higher in the pseudoautosomal region compared to the remainder of the strata on the X chromosome. We tested whether the higher density of transposable elements in the pseudoautosomal region is above what is seen across the autosomes. We randomly selected 2.5 Mb windows (the size of the pseudoautosomal region) from across the autosomes to generate a null distribution of transposable element density (measured as the proportion of nucleotides occupied by a transposable element). This analysis revealed the density observed in the pseudoautosomal region is not higher than in other recombining regions of the genome (10,000 permutations; $P = 0.135$). These results are reported on lines 434-443.

Reviewer #2: This ms by Peichel et al. reports the assembly of a largely non-recombining Y chromosome in fish. They used a combination of PacBio sequencing and Hi-C to obtain a high-quality assembly of a ~16 Mb Y chromosome in stickleback. Analyzing the Y sequence, they found its centromere, three evolutionary strata, an enrichment of haploinsufficient genes and an accumulation of testis-biased expression genes on the Y. Contrary to mammalian Y

chromosomes, they did not find any palindrome or ampliconic structure. They also proposed a candidate sex-determining gene (Amhy) based on work in other fish.

So far, largely non-recombining Y chromosomes have been sequenced in mammals only. The stickleback Y chromosome is thus a significant advance for the study of vertebrate sex chromosomes. The analyses presented in this ms are interesting and consistent with previous work on fragmented data on stickleback sex chromosomes. I think the most striking findings are 1) the finding of a candidate master sex-determining gene 2) the absence of palindromes or other ampliconic structures found in mammals. On this ground, I think that this ms merits a publication in Genome Biology, pending some modifications listed below.

1) They need to be clearer when presenting the literature and key definitions about sex chromosomes. Otherwise, many statements are too vague and prevent the reader to really understand the advance provided by this work. Some important references are missing. Also, a more precise definition of key terms will make the Discussion part clearer.

In the abstract and in the introduction, the author claimed that "Y chromosome sequence assemblies have only been generated across a handful of taxa with ancient sex chromosomes." and "Reference assemblies of young sex chromosomes are largely absent". Only papers on mammals and drosophila are cited.

I think the authors need to make a distinction between heteromorphic vs. homomorphic sex chromosomes or sex chromosomes with small vs. large non-recombining regions. They also need to be more precise than "old" or "young" to characterize sex chromosomes.

It is correct that only few Y chromosomes have been sequenced and assembled compared with the number of sequenced and assembled genomes. However, there are more of them than the mammalian and drosophila ones. Several homomorphic Y chromosomes have been sequenced and assembled (e.g. papaya, Wang et al. 2012; medaka, Kondo et al. 2006). The authors have only cited work on heteromorphic Y chromosomes. They should either specify it or cite work on homomorphic Y chromosomes as well. It is not to diminish their work, sequencing heteromorphic Y chromosomes is more difficult than sequencing homomorphic Y ones, but again I think it's better to be precise. Note that W, U and V chromosomes (also non-recombining as the Y) have also been sequenced but they can be neglected here...

Response: We thank the reviewer for raising these important points about the confusion created by our use of the terms “young” and “old” sex chromosomes and for providing many helpful references. We fully agree that it is important to be clear. Thus, in response, we have now shifted the focus from comparing young and old sex chromosomes, to comparing sex chromosomes at different stages of degeneration (e.g. Charlesworth et al. 2019). We further distinguish between homomorphic and heteromorphic sex chromosomes. We agree with the reviewer that the threespine stickleback Y chromosomes is likely at an intermediate stage of degeneration as it is already heteromorphic. Thus, we have mostly focused our discussions and references to comparisons of other heteromorphic sex chromosomes that have been studied at the sequence level. The Title, Abstract, Introduction, and Discussion have all been revised in response to these comments.

It is not the first study using PacBio to sequence Y chromosomes. Single molecule sequencing has a strong potential for getting high-quality assemblies of the Y chromosomes because of the long reads it generates. They cite a paper on the *Drosophila miranda* neoY, but PacBio has been used successfully in other species (e.g. gorilla, Tomaszewicz et al. 2016; mosquitoes, catfish; Bao et al. 2019). This should be acknowledged.

Response: Thank you for pointing us to these references. The gorilla (Tomaszkiewicz et al. 2016) and mosquito (Hall et al. 2016) references have been added. However, as we now specify that we are focusing on heteromorphic sex chromosomes, the catfish reference was not added because this Y chromosome is homomorphic.

Some of the homomorphic Y chromosomes that were sequenced are a few millions years old. The statement that no young Y chromosomes has been sequence before is not correct. Throughout the ms, the XY stickleback system is said to be "young" because it is less than 26 MY old. But the stickleback XY system is clearly heteromorphic. Young / old Y chromosomes: what does it mean?

Response: Please see our response above.

Several authors have warned against using absolute age to compare sex chromosome systems (Bachtrog 2011; Muyle et al. 2017). Species can have drastically different generation time and 1 MY for a mammal is not the same as 1 MY for a fly. To make fair comparison, one has to correct for the generation time. Another possibility is to use raw synonymous divergence instead of absolute times obtained from molecular clock.

In stickleback, the X-Y dS of the oldest stratum is 0.15 (see Figure 4 and Table 1 of this ms). It is similar to that of *Silene latifolia* (a plant XY heteromorphic system, also with several strata, e.g. Papadopoulou et al. 2015) and is higher than the X-Y dS of the strata 4 and 5 in primates. It is not that young and should be distinguished from incipient systems as the medaka XY one, for example.

Response: Although we still provide information about the absolute ages of various sex chromosome systems, where possible we also try to provide information about how this translates into number of generations. In the Discussion, we also now explicitly compare the rates of gene loss observed in sticklebacks to other systems with similar levels of synonymous divergence like *Silene latifolia*, *Drosophila miranda*, two species of *Rumex*, and strata 4 and 5 of the primate Y. See lines 563-579.

Referring to the steps of the evolution of heteromorphic sex chromosomes would be appropriate here. The stickleback system does not fit in with the first steps, it is more alike to the intermediate steps where heteromorphy is already visible (e.g Bachtrog 2013; Charlesworth et al. 2005).

Response: We now emphasize that the threespine stickleback is in the intermediate stage of degeneration.

I agree this problem is not specific to this manuscript and in the past every system younger than the mammalian Y (=180 MY old) would be called young, but this is not precise and should be avoided.

In line with this, I do not agree with this statement "The young sex chromosome of threespine stickleback is a useful model system to understand how the genetic architecture of sex-limited chromosomes initially evolves." I think the stickleback XY system is a useful system that provides insight on the intermediary steps of the evolution of heteromorphy, not the very first steps. To study the first steps, incipient sex chromosomes with tiny non-recombining regions are more relevant (e.g. Charlesworth 2019).

In the Discussion: "Despite the young age of the threespine stickleback Y chromosome relative to mammals" is too vague... See my comments above to rephrase it.

Response: Please see our responses above. All of these comments have been addressed through revision of the Title, Abstract, Introduction, and Discussion.

2) I would welcome a bit more details on how the assembly was done. It is indicated in the ms that the individual that was sequenced was heterozygous. In this case, Falcon and Falcon unzip are known to perform really well. It would be nice to have a justification of why Canu was used here instead. Also, the way the X and Y chromosomes were assembled separately is not entirely clear. A high-quality assembly of the Y should indeed not be a chimeric assembly with bits of X and Y but this is not easy when using DNA from a male genome and not from flow sorted Y chromosome or Y-probe screened BAC libraries as in some of the previous work cited in the introduction. It would be nice to include some tests showing that the X and the Y here are not chimeric. The result of the mapping of reads from a female genome on the X and Y references could be interesting. Identifying of Y-linked SNPs from multimale - multifemale comparison and checking that these are present on the Y reference only could be too.

Response: Canu was chosen as the assembler because it performed substantially faster than Falcon and Falcon unzip, which allowed us to rerun the assembly pipeline multiple times to test different parameter values. We validated the completed assembly with multiple independent data sources (i.e. Sanger-sequenced BAC inserts were largely concordant with the assembly, the centromere was correctly positioned, and the overall ordering of cytogenetic markers was concordant), which indicated at a broad-scale that the Y chromosome assembly was not a chimera of the X and Y chromosome. However, the reviewer brings up a valid point that there may be smaller regions of the Y chromosome that are a chimera of both sex chromosomes. This would result if some of the Canu assembled contigs contained PacBio raw reads from both the X and Y chromosome and would occur in regions of the X and Y that share very high sequence similarity. In order to test for fine-scale evidence of chimeric regions, we aligned Illumina short-read sequences from female and male threespine stickleback fish from three different populations and looked at overall sequence alignment to the X and Y chromosomes, relative to the autosomes. If there are chimeric regions of the Y chromosome, reads from females should align equally well to the X and Y in these regions, resulting in 0.5X coverage on the Y. If the Y is largely not chimeric, female reads will only align to the X chromosome, resulting in no coverage on the Y chromosome. As expected, in all three populations the median read coverage relative to autosomes in females was 0.0x on the Y chromosome and was around 1.0X on the X chromosome (Supplemental Figures 5, 6, and 7). In males the X and Y chromosomes had 0.5X coverage relative to autosomes, indicating Illumina short-reads are correctly partitioned between the sex chromosomes. At a fine scale, there was some variation in coverage throughout the X chromosome, which could be caused by some reads cross-aligning to both sex chromosomes in highly homologous regions (Supplemental Figures 5, 6, and 7). Indeed, these few regions seemed to be enriched in the youngest strata of the sex chromosomes where sequence homology is highest. It is challenging to disentangle whether these few regions represent true chimeras or reflect accurate assembly of highly homologous regions. We include these results in a new section entitled "Short-read sequencing correctly aligns to both X and Y chromosomes" on lines 271-285 and in the discussion section on lines 492-503.

We apologize that our description of the X and Y assembly was not completely clear. We have now included a statement in the “Hi-C proximity guided scaffolding” section of the methods (lines 688-693) to clearly outline what sets of contigs were used for the Y chromosome assembly: “Hi-C proximity guided scaffolding could not accurately scaffold the X and Y chromosomes from a combined set of contigs. To simplify the scaffolding process, we separated putative X- and Y-linked contigs from the genome-wide set of contigs. Putative Y-linked contigs were identified as: (1) contigs that aligned to the reference X chromosome, but with higher sequence divergence; and (2) contigs that only partially aligned or did not align at all to the revised female reference genome.”

3) It is not clear to me how the PAR was identified here. This should be explained more precisely. Also, I would add the PAR in Figure 4.

Response: We now include a statement in the methods section describing how the PAR boundary was defined (lines 848-852): “The pseudoautosomal region boundary was set at 2.5 Mb, as previously defined by patterns of molecular divergence between the X and Y chromosomes (White et al. 2015) and through genetic linkage maps (Peichel et al. 2004; Roesti et al. 2013; Sardell et al. 2018).”

The PAR is not in Figure 4 because this region is missing from the Y chromosome assembly, and we cannot estimate dS between gametologs. Rather than include the PAR in Figure 4 without data points, we have indicated in the figure legend that the PAR is not shown.

4) I think it would be nice to include an analysis of Y degeneration here. What is the extent of gene loss on the Y chromosome of stickleback? It would be really interesting to compare that number with other comparable systems. The best option would be to use a fish with a different sex chromosome system as an outgroup and to infer gene loss (and gain) on the X and Y separately in stickleback. An easier option (assuming the gene content of the X has not changed too much) is to compare directly X and Y chromosomes' gene content in stickleback.

Response: A direct comparison of the X and Y gene content, showing that only 44.1% of all annotated genes on the X chromosome are present on the Y chromosome, was already provided in the manuscript (see lines 339-340; Table 1). Thus, we have not done any additional analyses using an outgroup species. However, we have now emphasized this result by adding a Discussion that explicitly compares the rates of gene loss observed in threespine stickleback to other systems with similar levels of synonymous divergence like *Silene latifolia*, *Drosophila miranda*, two species of *Rumex*, and strata 4 and 5 of the primate Y. See lines 563-579.

5) They found that TEs are abundant in the PAR. This is not expected. TEs are expected to accumulate in non-recombining parts of the sex chromosomes. PARs are typically highly recombining and are not expected to contain many repeats. This should be discussed (e.g. is recombination rate low in the stickleback PAR?). A similar pattern has been found in the brown alga *ectocarpus* (Luthringer et al. 2015).

Response: We agree with the reviewer that the high density of transposable elements in the pseudoautosomal region was surprising, given that the accumulation of these elements should be slower in regions of the genome that are recombining. Thus, based on the reviewer's comment, we decided to test whether the abundance of TEs on the

PAR was greater than in autosomal regions of the genome. To do so, we randomly selected 2.5 Mb windows (the size of the pseudoautosomal region) from across the autosomes to generate a null distribution of transposable element density (measured as the proportion of nucleotides occupied by a transposable element). This analysis revealed the density observed in the pseudoautosomal region is not higher than other recombining regions of the genome (10,000 permutations; $P = 0.135$). These results are reported on lines 434-443.

6) Their proposition that *Amhy* is the master sex-determining gene in stickleback is interesting. Of course, genetic validation and functional studies are required to really test this idea. One thing that could be checked here is the dS value of the *AmhY* with its autosomal parent. This dS should be consistent with the mean dS of the stratum 1 (or at least fall in the range of dS values of that stratum) if the evolution of *Amhy* as a master sex-determining gene is associated with the evolution of stratum 1.

Response: This is an excellent idea. We used PAML to estimate dS of *AmhY* compared with its autosomal parent. dS was 0.477, which is within the range of values for genes in stratum one (median dS: 0.155; interquartile range: 0.081 – 0.611). This value is not within the interquartile range of genes in stratum two (interquartile range: 0.027 – 0.075) or stratum three (interquartile range: 0.026 – 0.052). We report these results on lines 456-460 and in the methods on lines 946-947.

7) p 22, there is a discussion on the differences between dS values estimated from Illumina assemblies and from PacBio assemblies. dS values are found higher in the latter than in the former. It is well known that PacBio has a high sequencing error rate. One immediately wonders whether the differences in dS result from this and is a mere methodological artefact. How much the auto-correction of the PacBio contigs and the polishing were successful should probably be discussed here to rule out his possibility.

Response: The genome assembly pipeline we used incorporates error correction at two stages. During the first stage, the Canu assembler uses overlap information to correct raw reads. In the second stage after contigs were generated, there was a polishing step using arrow. This program uses alignment of raw reads to the contigs in order to generate accurate consensus calls. Our assembly had approximately 37.6X coverage each for the X and Y chromosomes (75.25X genome-wide) for the error correction steps, which should allow for very high accuracy in our consensus contigs. To determine whether the synonymous divergence we observed between the X and Y chromosomes could have been driven by residual sequencing error in the contigs, we aligned male Illumina short-read paired-end sequences from three separate populations to the reference Y chromosome assembly. If the Y reference assembly has a high error rate, we would expect increased nucleotide divergence between the Y reference assembly and the Illumina sequenced Y chromosome. Instead, we found nucleotide divergence between the Y reference and the three populations was much lower than population divergence between autosomes. The lowest divergence observed was in a comparison with a male isolated from the same population as the male used for the reference Y chromosome (Paxton Lake benthic population). These results indicate the high synonymous divergence we observe between the X and Y chromosomes cannot be from sequencing error. Interestingly, this analysis also revealed that threespine stickleback fish have low within-species polymorphism on the Y chromosome, consistent with reduced diversity on Y chromosomes in other species (Wilson-Sayres et al. 2014; Hough et al. 2017; Wilson-Sayres 2018). This is an interesting result to explore in the future!

These results are outlined in the results on lines 352-362 and in Supplemental Table 2 and are discussed on lines 492-503. The methods are outlined on lines 775-801.

8) The proposed scenario for the formation of the 3 strata (shown in Figure S9) does not make sense to me. As far as I understand the figure, it looks like in step 2, there is a recombining region between strata 1 and 2 (in addition to the PAR at the tip of the sex chromosomes). This implies that strata 1 and 2 would not be genetically linked at this stage. I am not saying that is impossible. Actually, it would be very interesting. Theory says indeed that we should not observe such an event. In the current model for the evolution of heteromorphy, the non-recombining region on the Y tends to grow over time by genetically linking fragments of the Y to the male-determining gene(s). It seems to me that other scenarios of the chromosomal rearrangements between X and Y chromosomes should be explored and discussed. What about an inversion on the X chromosome encompassing strata 2 and 3 following the formation of strata 3? This has happened on the bird Z chromosomes (Zhou et al. 2014; Yazdi & Ellegren 2018). Using an outgroup (with a different sex chromosome system) could help pinpoint the inversions that took place between the stickleback X and Y chromosomes.

Response: Indeed, we have also puzzled over this result, as it is inconsistent with theoretical predictions! The suggestion that there has been an inversion on the X chromosome is interesting and is supported by a comparison of the genome assemblies of the threespine stickleback and ninespine stickleback. The X chromosome and its corresponding autosome in ninespine stickleback have multiple and complex rearrangements. Disentangling whether these rearrangements have occurred on the threespine or the ninespine lineage will require genome assemblies from additional stickleback species (which are currently underway). Thus, we cannot currently pinpoint the inversions that have occurred between the stickleback X and Y chromosomes, but we have now added these points to the Discussion. See lines 519-531.

9) The analysis on testis-biased gene accumulation is interesting but I am not sure that the approach they used is the best suited. They compared the accumulation of tissue-specific genes on the Y comparing three tissues: testis, brain and liver. Their results, however, could be explained by a genome-wide excess of testis-biased genes compared to other tissues and not a Y-specific excess. To know for sure, one would need to compare all chromosomes. The studies that have established an accumulation of sex-biased genes on the sex chromosomes have done so (reviewed in Parsch & Ellegren 2007, 2013; and see for example Assis & Bachtrog 2012). Here, I think looking at the distribution of testis-biased gene in the genome would be required to back up the claim that the Y is enriched in those genes.

Response: We agree with the reviewer that genome-wide analyses would be necessary if we were testing for an enrichment of testis-biased genes on the Y chromosome. However, our analysis is specifically looking at two categories of genes on the Y chromosome that are predicted to have male-biased functions: genes that have moved onto the Y chromosome from the autosomes through translocation (Figure 6, dark grey) and genes that have X-linked gametologs, but have amplified on the Y chromosome through duplication (Figure 6, light grey). We test whether these two categories of genes exhibit greater testis-biased expression relative to the remainder of ancestral, single copy genes on the Y-chromosome (Figure 6, black). In the analysis, we are not showing the Y chromosome is enriched overall for testis-biased genes (in fact, the single copy ancestral genes as a whole do not show testis-biased expression in two of the tissue

comparisons), but we show that these newly acquired gene families on the Y chromosome do exhibit greater testis-biased expression relative to the remainder of the single-copy Y-linked genes. This pattern is repeated in three different tissue comparisons. To clarify our analysis, we have removed all phrasing in the results and discussion sections that used the word “enrichment.”

10) As mentioned above, the stickleback XY and the *Silene latifolia* XY systems have similar level of heteromorphy and age ($dS \sim 0.15$, 3 strata, significant differences in X/Y size). It is noteworthy that the stickleback Y chromosome is smaller than the X, when the *S. latifolia* Y is larger than the X (also observed in *Coccinia grandis*, another plant heteromorphic system similar to *S. latifolia*, Sousa et al. 2013). This raises the possibility that the evolution of Y size may be different in animals and plants, an interesting idea.

Response: This is certainly an interesting idea, although the data are currently limited. Despite loss of over half of the genes on the threespine stickleback Y chromosome, our assembly and cytogenetic data suggest that the Y is only slightly smaller than the X, likely due to the accumulation of TEs and acquisition of novel genes on the Y chromosome. A similar loss of 70% of genes without shrinking of the Y chromosome relative to the X has recently been found in the plant *Cannabis sativa* (Prentout et al. (2020) *Genome Research* 30: 164-172). In both of these cases, it is unknown whether there was an earlier stage in which the Y expanded to be larger than the X and later shrank, or whether the rate of TE accumulation is slower in threespine stickleback and *C. sativa* relative to *S. latifolia* or *C. grandis*. Given the paucity of data for a proper comparison as well as the current length of our manuscript, we prefer to refrain from adding a discussion of this interesting speculation.

Second round of review

Reviewer 2

The authors have done a nice job in replying to my comments. Almost all of them have been addressed in a satisfactory way.

I am still not convinced at all with their model for the strata formation (cf my comment #8 and their reply). They agreed with my suggestion that an inversion on the X chromosome may explain the strata order that they observed. They also agreed that using an outgroup is needed to identify the chromosomal rearrangements that took place between the X and the Y. Their model is thus highly speculative and I think that it weakens the ms. I think that they should either (1) present different possible scenarios in the text (their scenario + another one with an inversion on the X) and make a new figure S13, or (2) say that nothing cannot be concluded without an outgroup about strata formation and remove figure S13.

Authors Response

We have addressed the final concern of reviewer two by removing the discussion of how the inversions formed on the Y chromosome. We have also removed Supplemental Figure 13. We agree that our model was speculative without an outgroup and we think removing this discussion strengthens the overall manuscript, as suggested by the reviewer.